# Correction of the molecular phenotype of X-linked Dystonia-Parkinsonism reveals a non-canonical function of BRD4

Simona Capponi[1,2], Sandra Ehret[1], Zeynep Camgöz[1,2], Fabian Gather [3], Christine A. Vaine[4,5], Ezgi Özyerli-Göknar [1,2], Marie Follo [6], D. Cristopher Bragg[4,5], Tanja Vogel [3] & H. Th. Marc Timmers [1,2] ✉

Transcription and mRNA processing are tightly coupled regulatory layers on gene expression, and their perturbations underly human disorders. X-linked Dystonia-Parkinsonism (XDP) is a unique example of a human disease connecting aberrant mRNA processing and the basal transcription machinery. XDP is a rare, monogenic fatal neurodegenerative disorder, and a limited understanding of the underlying molecular mechanisms hinders the development of effective therapies. In this study, we show that depletion of BRD4, a chromatin reader known for its role in transcriptional pausing, rescues the XDP molecular signature. Unexpectedly, this effect is independent of the canonical coactivator role of BRD4. We demonstrate that the XDP-SVA induces intronic premature cleavage and polyadenylation within the *TAF1* locus, and that BRD4 depletion bypasses this premature termination checkpoint. These findings reveal new dimensions of BRD4 activity beyond transcription pause release and suggest modulation of mRNA processing as a therapeutic strategy for XDP.

The fidelity of gene transcription is critical for supporting development, maintaining cellular identity, and ensuring tissue homeostasis. While RNA polymerase II (pol II) drives the synthesis of messenger RNAs (mRNAs), its activity is tightly coupled to co-transcriptional processes including splicing, cleavage and polyadenylation, that collectively shape transcript maturation. Disruption of any of these steps can compromise mRNA integrity and alter gene output with broad consequences for cell function, resulting in human diseases.

Among the many challenges to transcriptional integrity are repetitive elements, which comprise nearly half of the human genome. One impactful class is the SINE-VNTR-Alu (SVA) elements, primate-specific retrotransposons active during early development before being epigenetically silenced[1]. Depending on their genomic context, SVAs can interfere with transcriptional and co-transcriptional processes. The pathological impact of SVAs is illustrated in neutral lipid storage disease with myopathy, where an SVA insertion in exon 3 of the *patatin-like phospholipase domain containing 2* gene promotes gene silencing and lipid accumulation in muscles[2]. In contrast, Fukuyama congenital muscular dystrophy exemplifies the co-transcriptional effects of SVAs, acting as an exon trap, disrupting splicing and producing a truncated, dysfunctional Fukutin protein[3].

While these examples illustrate how SVA insertions can interfere with gene expression through disruptions of chromatin regulation or splicing, the situation in X-linked Dystonia-Parkinsonism (XDP, OMIM:

[1]Department of Urology, Medical Centre, Faculty of Medicine, University of Freiburg, Freiburg, Germany. [2]German Cancer Consortium (DKTK), partner site Freiburg, German Cancer Research Center (DKFZ), Heidelberg, Germany. [3]Institute for Anatomy and Cell Biology, Department of Molecular Embryology, Faculty of Medicine, Albert-Ludwigs-University Freiburg, Freiburg, Germany. [4]Department of Neurology, Massachusetts General Hospital and Harvard Medical School, Boston, MA, USA. [5]Collaborative Center for X-linked Dystonia-Parkinsonism, Massachusetts General Hospital, Charlestown, MA, USA. [6]Department of Medicine I, Lighthouse Core Facility, Medical Center - University of Freiburg, Faculty of Medicine, University of Freiburg, Freiburg, Germany. ✉e-mail: m.timmers@dkfz-heidelberg.de

314250) appears markedly more complex. XDP is a rare, adult-onset neurodegenerative disorder predominantly affecting males of Filipino descent[4]. It is characterized by a combination of progressive dystonia and Parkinsonism, with neuropathology primarily involving degeneration of the striatum[5], a brain region which coordinate voluntary movements. The phenotype results from the insertion of an SVA in intron 32 of the *TBP-associated factor 1* (*TAF1*, OMIM: 313650) gene[6]. *TAF1* encodes the largest subunit of the Transcription Factor IID (TFIID) complex, which plays a central role in initiating gene transcription by recruiting pol II to core promoter elements[7]. This SVA insertion results in aberrant processing of *TAF1* intron 32 including intron retention and activation of cryptic splice acceptor sites located upstream of the XDP-SVA. These lead to a measurable (~20%) reduction in full-length *TAF1* mRNA which affects a subset of downstream genes critical for neuronal identity and function[8]. These alterations collectively define the XDP molecular signature. Yet, despite its consistency across patient-derived models, the precise mechanism by which the XDP-SVA disrupts *TAF1* processing, and whether this defect can be reversed, remains unresolved.

In this study, we employed an integrated approach combining a fluorescent reporter system with unbiased chemical screening to probe the basis of *TAF1* processing defects and evaluate strategies for molecular correction. Through this approach, we identified depletion of Bromodomain Containing 4 (BRD4) as a potent modulator of the aberrant *TAF1* mRNA misprocessing. Further dissection revealed that this effect is independent of the canonical function of BRD4 in chromatin engagement and promoter-proximal pause-release[9] and that it instead relies on its non-canonical role in 3′ mRNA processing. Our work demonstrates that the XDP-SVA triggers activation of cryptic exons resulting from mis-splicing and premature cleavage and polyadenylation (PCPA) within *TAF1* intron 32, which we collectively define as defects in mRNA processing. We show that BRD4 depletion suppresses this defect to restore production of full-length *TAF1* mRNAs. These findings provide mechanistic insight into XDP pathogenesis and support a model in which dysregulated transcription termination represents a critical vulnerability that may be exploitable in future therapeutic strategies.

## Results

### An XDP-SVA *TAF1* reporter recapitulates the XDP molecular signature

To investigate how the XDP-SVA insertion affects *TAF1* processing, we engineered a minigene (MG) fluorescent dual-color reporter that models the genomic context of the disease (MG-SVA). The construct spans the human *TAF1* locus from exon 31 to 35 and incorporates the XDP-SVA at its native landing position within intron 32, which has been reconstructed using the native 5′ and 3′ regions and retaining ~1 kb of intronic sequence flanking the XDP-SVA, including the previously described cryptic exon i32[8,10,11]. The construct features a dual-fluorophore design, where *TAF1* exons are fused in-frame at the 5′ end with GFP and at the 3′ with mKATE2 to monitor transcriptional activity across the XDP-SVA insertion. The construct is paired with a wild-type control, which shares the same molecular markup, but lacks the XDP-SVA (Fig. 1a).

When introduced into HeLa cells, the XDP-SVA-containing *TAF1* reporter reproduces the core molecular features of XDP. Compared to the wild-type construct, the presence of the XDP-SVA causes a marked reduction in reporter-specific mRNA output with a progressive decrease expression, that is more pronounced for exons downstream of the XDP-SVA (Fig. 1b). This defect is accompanied by robust activation of the cryptic intronic exon i32 located upstream of the XDP-SVA (Fig. 1c).

To investigate the link between transcriptional imbalance across the XDP-SVA (Fig. 1d) and incorporation of the intronic exon i32, we analyzed the structure of generated transcripts. We performed 3′ Rapid Amplification of cDNA Ends (3′ RACE) coupled with long-read sequencing to capture the full spectrum of stable, polyadenylated minigene-specific transcript isoforms. While transcript isoforms derived from the wild-type minigene show a robust usage of exon 33 splicing acceptor site (Fig.1e) and full-length minigene mRNAs represent the vast majority of detected isoforms (Fig. 1f), the presence of the XDP-SVA altered this pattern. Specifically, exon 32 was mainly linked to the cryptic splicing acceptor site of intronic exon i32 (Fig. 1e). Isoform phasing confirmed that the incorporation of intronic exon i32 was coupled with premature mRNA cleavage and polyadenylation (Fig.1f and Supplementary Fig. 1a-d). This data demonstrates that i32 does not splice onward to downstream exons but, instead defines the endpoint of the transcript. This observation was further supported at protein level. The wild-type *TAF1* minigene produced a ~70-kDa protein corresponding to the full-length product. This was detected by both a GFP antibody and a TAF1-specific antibody, which targets the junction between exons 34 and 35 (Fig. 1a and ref. 12). In contrast, the XDP-SVA-containing reporter yielded reduced levels of full-length protein and additional truncated species, that were GFP-positive but lacked reactivity to the TAF1 antibody (Fig. 1g) consistent with translation from transcripts terminating within intron 32. The observed imbalanced transcription across the XDP-SVA was also reflected in fluorophore expression: while GFP remained detectable, mKATE2 fluorescence was markedly reduced (Fig. 1h).

Taken together, these data confirm that the *TAF1* minigene reporter recapitulates the molecular signature of XDP. Misprocessing of the reporter was consistently observed across multiple cell lines (Supplementary Fig. 1e), as well when the CMV promoter was replaced by *TAF1* promoter sequences (Supplementary Fig. 1f-i). Notably, while patient lines typically exhibit ~20% downregulation of full-length *TAF1*[8], the CMV-driven reporter amplifies this effect providing a direct, live-cell readout of transcription integrity across the XDP-SVA. We reasoned that this system, by enhancing the dynamic range of the phenotype, could be applied as a functional screening platform. In this view, molecules able restore mKATE2 fluorescence would reveal molecular players and cellular pathways underlying the XDP-SVA-induced transcriptional imbalance over the *TAF1* locus.

### BET depletion rescues the XDP molecular signature

To test this idea, the dual-color XDP-SVA *TAF1* reporter driven by the CMV promoter provided a sensitive platform for small molecule screening based on automated microscopy to identify compounds capable of increasing mKATE2 over GFP fluorescence. We screened the FREpi library (Supplementary Fig. 2a to 2c and Supplementary Table 1), a custom-designed, curated collection of over 500 molecules enriched for targets related to transcriptional and epigenetic regulation. The screening yielded several positive hits, of which the majority clustered within a distinct chemical space: molecular degraders of the Bromodomain and Extra-Terminal domain (BET) protein family including BRD2, BRD3, BRD4 and the testis-specific BRDT (Fig. 2a and Supplementary Fig. 2d). The compounds ARV-771, MZ1, dBET6 and ZXH 3-26 belong to the class of PROTACs (PROteolysis-TArgeting Chimeras), which induce selective degradation of target proteins by directing them to the ubiquitin-proteasome system[13]. These molecules are bivalent: one part binds BET proteins via the small molecule JQ1, a known BET-bromodomain inhibitor, while the other part recruits an E3 ubiquitin ligase, promoting poly-ubiquitination and subsequent proteasomal degradation of BET targets[14].

Depletion of BET proteins acted on the *TAF1* locus in an SVA-dependent manner. In cells containing the wild-type reporter, BET degraders induced a mild increase in overall minigene transcription without affecting the mKATE2-to-GFP ratio (Fig. 2b). In contrast, in the presence of the XDP-SVA, the compounds significantly increased the mKATE2-to-GFP signal to wild-type levels in a dose-dependent manner (Fig. 2b–c and Supplementary Fig. 3a–c). Transcript analysis confirmed

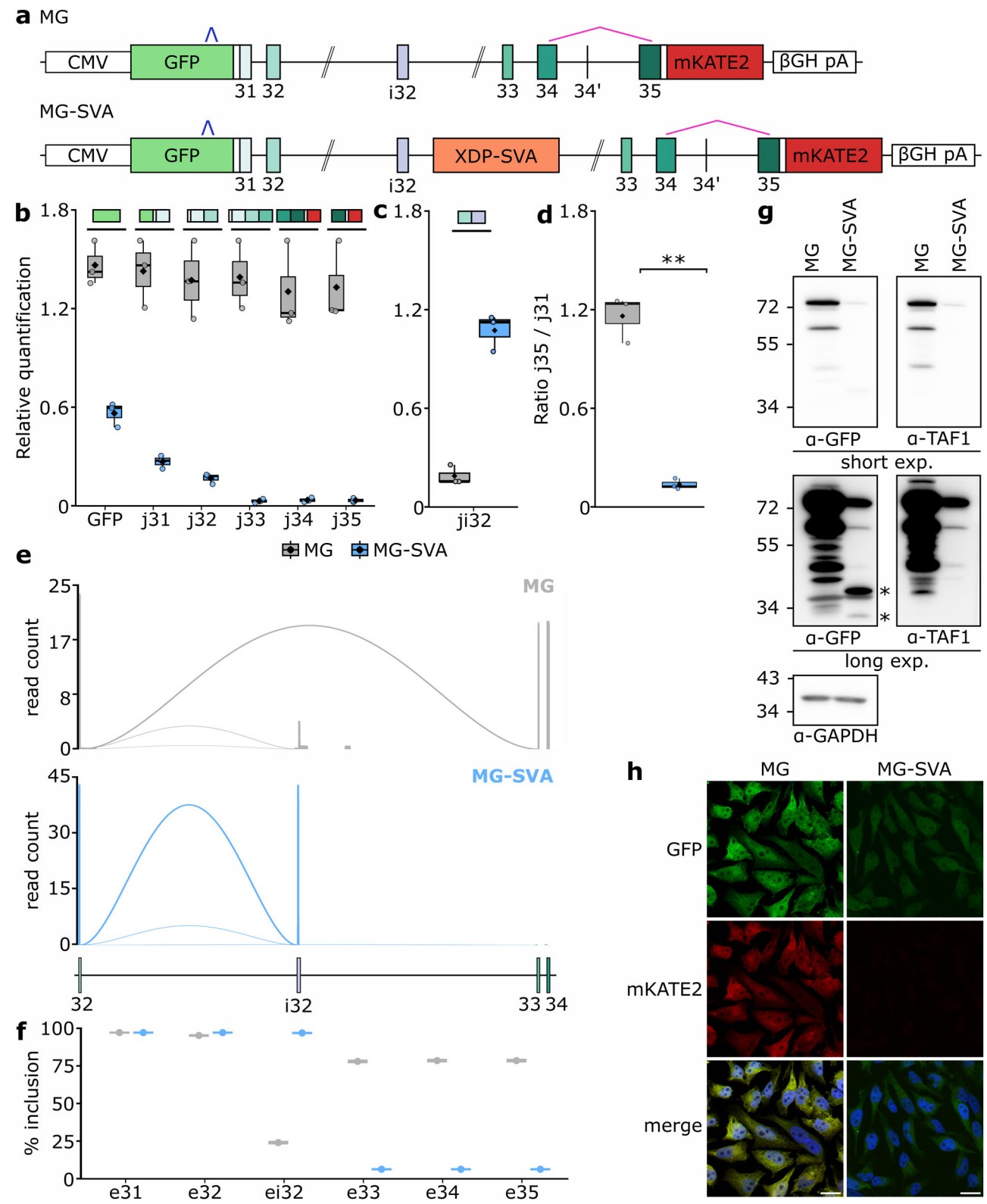

XDP-SVA-dependent, improved processivity across the *TAF1* locus following BET depletion with increased splice acceptor usage (Fig. 2d–e and Supplementary Fig. 3e–g) and expression of exon 33 (Supplementary Fig. 3d). This dynamic suggests a model in which the XDP-SVA insertion triggers defective mRNA processing by enhancing premature termination checkpoint at cryptic exon i32. In this context, BET depletion modulates this process thereby promoting transcriptional readthrough past this point and increasing full-length transcript production.

## BRD4 regulates 3′ mRNA processing independently of its bromodomains

The BET protein family plays key roles in transcriptional regulation through chromatin engagement. All BET proteins, BRD2, BRD3, BRD4

**Fig. 1 | The XDP-SVA dual-color *TAF1* minigene reporter recapitulates the XDP molecular signature. a** Schematic representation of the MG and MG-SVA minigene reporters. Exons are shown as boxes and numbered; the XDP-SVA insertion is indicated. The epitope recognized by the TAF1 antibody is indicated in pink; the GFP antibody is shown for reference in blue. Relative abundance of splicing junctions for canonical exons (**b**) and i32 (**c**) quantified by RT-qPCR for MG and MG-SVA. **d** Ratio of exon 35-mKATE2 to linker-exon 31 RT-qPCR products (*p* = 0.0046). **e** Sashimi plot depicting splicing acceptors usage from exon 32 splicing donor site. Read count is depicted as read counts/1000. **f** Dot plot indicating the percentage of

incorporation of the different minigene exons for the identified transcript isoforms. Error bars indicate Wilson 95% confidence intervals. **g** Immunoblot of *TAF1* minigene-derived protein products using GFP and TAF1-specific antibodies. Truncated products in XDP-SVA-expressing cells are indicated with asterisks (*). Molecular weight is expressed in kDa. **h** Representative fluorescence microscopy images showing GFP and mKATE2 expression in MG- or MG-SVA-containing cells. Scale bars, 20 µm. Box plots (**b**–**d**) are generated with *n* = 3 biological replicates. Dot plot (**f**) is generated with *n* = 1. Source data are provided as a Source Data file.

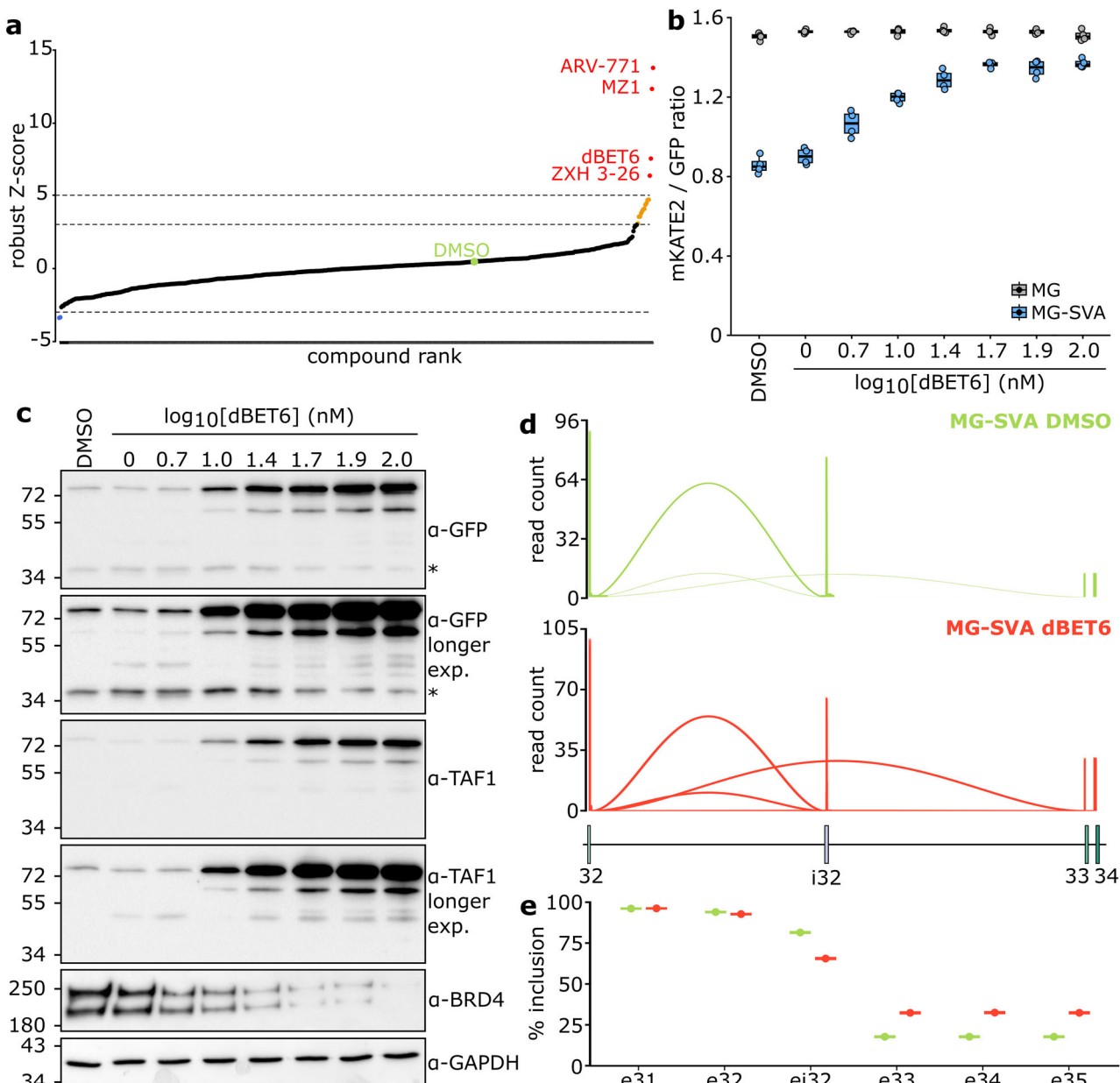

**Fig. 2 | BET depletion rescues the XDP molecular signature. a** Positive hits from the FREpi library screen. Compounds with a robust Z-score (rZ) > 5 are shown in red; those between 3 and 5 in orange; non-significant hits (−3 <rZ <3) in black; negative hits (<−3) in blue. **b** Dose-dependent effect of dBET6 on mKATE2/GFP ratio of fluorescence signals in MG- and MG-SVA-containing cell lines. Box plots generated with *n* = 4 technical replicates. **c** Dose-dependent effect of dBET6 on minigene-derived proteins. Truncated products are marked with an asterisk (*). BRD4 depletion is shown and GAPDH serves as a BET depletion-insensitive loading

control. Molecular weight is expressed in kDa. **d** Sashimi plots depicting the usage of i32 or exon 33 acceptor sites upon DMSO (top panel, in green) or dBET6 (bottom panel, in red) treatment. Read count is depicted as read counts/1000. **e** Dot plot (*n* = 1) indicating the percentage of incorporation of the different minigene exons for the identified transcript isoforms upon DMSO (green) or dBET6 (red) treatment. Error bars indicate Wilson 95% confidence intervals. Source data are provided as a Source Data file.

and the testis-specific BRDT, contain two conserved N-terminal bromodomains, that bind acetylated lysines on histone tails, anchoring them to active chromatin regions[15]. Bromodomain inhibitors such as JQ1, which competitively block BET protein binding to acetylated chromatin[16], have been widely used to probe BET-dependent transcriptional regulation. Since JQ1 serves as BET-bromodomain binder in the BET PROTAC molecules, we first investigated whether inhibition of BET-bromodomains alone would be sufficient to modulate the transcriptional output of the XDP-SVA minigene reporter. Interestingly, treatment with JQ1, as well as with other structurally distinct bromodomain inhibitors (OTX-015 and I-BET151), failed to restore transcription across the XDP-SVA (Fig. 3a–c; Supplementary Fig. 4a–c). This suggests that BET-bromodomains are not implicated in 3' mRNA processing at the *TAF1* locus, and that this function possibly relies on a different protein domain.

Since BET proteins share a high degree of structural similarity and are often functionally redundant[17], we next determined which BET family member mediates the rescue of *TAF1* mRNA misprocessing. Selective depletion of *BRD2*, *BRD3*, or *BRD4* using siRNAs identified *BRD4* as the key player as its specific knockdown rescued substantially the XDP-SVA-induced transcriptional imbalance. *BRD2* depletion led to a modest but detectable effect, suggesting it may contribute, whereas *BRD3* knockdown had no measurable impact (Fig. 3d and Supplementary Fig. 4d). These results point to BRD4 as the primary effector of the observed rescue.

Together with the well-established role of BRD4 in regulating chromatin engagement and pol II pausing at gene promoters, recent studies have implicated BRD4 in 3' mRNA processing. In particular, dBET6-mediated BRD4 degradation has been shown to induce transcriptional readthrough at a subset of genes by promoting skipping of canonical polyadenylation signals leading to extended transcripts[18]. We reasoned, that while polyA signal skipping is generally disruptive, it could be beneficial at the *TAF1* locus, where the presence of the XDP-SVA triggers intronic splicing and PCPA in the proximity of exon i32. In this context, BRD4 depletion might lead to bypass intronic premature mRNA cleavage restoring readthrough and full-length transcript production. To explore this possibility, we analysed chromatin-associated reporter-specific RNA, which confirmed a marked increase in nascent transcripts detected downstream of the XDP-SVA upon dBET6 treatment (Fig. 3e and Supplementary Fig. 4e) supporting a model in which BRD4, directly or indirectly, alleviates mRNA misprocessing at the XDP-SVA insertion site.

We next sought to determine which domain of BRD4 mediates this effect. Beyond its two bromodomains, BRD4 harbors an extraterminal (ET) domain, which facilitates protein-protein interactions with transcriptional and chromatin regulators[19], and a C-terminal domain (CTD), which interacts with pol II and the transcriptional pause-release complex P-TEFb (Positive Transcription Elongation Factor b) composed of cyclin-dependent kinase 9 (CDK9) and Cyclin T1/T2[20]. Therefore, we generated a series of BRD4 deletion constructs (Fig. 3f and Supplementary Fig. 4f) and assessed their ability to restore XDP-SVA sensitivity upon endogenous BRD4 depletion. In this assay, endogenous BRD4 is depleted by dBET6, which relieves premature termination and restores proper transcript output, but the reintroduction of dBET6-insensitive BRD4 fragments would re-establish premature termination. Notably, while reintroduction of the N-terminal portion of BRD4 truncated either at the bromodomains or at the ET domain did not impact transcriptional readthrough, expression of the BRD4 CTD alone was sufficient to re-establish transcriptional repression across the XDP-SVA (Fig. 3g). This effect was independent of the known CTD interaction with P-TEFb, as CDK9 depletion failed to relieve the XDP-SVA-induced block (Supplementary Fig. 5).

Taken together, these results point to BRD4 CTD as a potential candidate region involved in regulating 3' mRNA processing at the XDP-SVA locus.

## XDP cerebral organoids show impaired neuroepithelial organization

The *TAF1* reporter assay combined with a comprehensive compound screening provided mechanistic insights linking the XDP molecular signature to alteration of mRNA processing. To evaluate the relevance of these findings in a disease-relevant model we generated cerebral organoids (COs) from XDP patient-derived induced pluripotent stem cells (iPSCs). These 3D structures recapitulate early stages of brain development in vitro[21] and recent studies in neurodegenerative conditions mirroring XDP including Parkinson's and Huntington's disease[22,23] demonstrated that, despite the early developmental stage, COs can be used as complex neuronal systems to recapitulate disease-specific molecular changes. In contrast to monolayer neuronal cultures, which offer limited cellular diversity and reduced transcriptional complexity, COs provide an organized three-dimensional neuronal environment, which more closely reflects brain architecture, allowing to assess the persistence and modulation of XDP-associated RNA processing defects.

XDP COs showed a consistent reduction in organoid size compared to controls at early differentiation stages (day 40 of COs maturation) alongside a decreased yield of COs per differentiation experiment (Supplementary Fig. 6a and 6b). Immunostaining for pan-neuronal (Tubulin beta 3 - TUJ1) and neural progenitor (Paired box 6 - PAX6) markers revealed morphological differences in neuroepithelial organization. Control COs displayed typical radial arrangements of neural progenitors, forming well-defined, circular rosette structures surrounding ventricle-like cavities (Fig. 4a). In contrast, XDP COs exhibited distorted architecture with expanded and irregular PAX6⁺ zones. These regions lacked the expected radial symmetry and, in some cases containing invaginated tissue within the ventricular cavity (Fig. 4b and Supplementary Fig. 6c–e). Morphometric analysis confirmed these observations: the circularity of rosettes was significantly reduced in XDP COs (Fig. 4c), while both the overall area of the ventricular zone and the thickness of the PAX6⁺ progenitor layer were increased (Fig. 4d–e). These findings point to early defects in progenitor number and their spatial organization in this organoid model.

Given that the CO model recapitulates the radial architecture of the developing human cortex with PAX6⁺ radial glia lining the ventricular zone and T-box brain protein 2-positive (TBR2⁺) intermediate progenitors forming a surrounding subventricular layer[24], we next examined whether the expanded PAX6⁺ domain in XDP COs was accompanied by alterations in subsequent progenitor populations. Indeed, immunostaining of TBR2 revealed a notable increased number of intermediate progenitor cells in XDP COs compared to controls, in which the TBR2 signal was markedly less pronounced (Fig. 4f). The expansion of both ventricular and subventricular layers suggests an imbalance in early neurogenesis potentially reflecting altered progenitor fate decisions or impaired transitions along the radial glial-intermediate progenitor-neuron trajectory.

Premature neurogenesis can trigger apoptosis in newly born neurons[25]. To determine whether aberrant tissue organization in XDP COs is accompanied by increased cell death, we assessed apoptotic markers in whole-organoid lysates. Immunoblot analysis revealed elevated cleaved Poly(ADP-ribose) polymerase (PARP) and cleaved (17 kDa) to full-length (34 kDa) caspase-3 ratio (Supplementary Fig. 6f) indicating increased apoptosis during early development consistent with enhanced apoptosis reported in other XDP model systems[26,27]. Increased cell death may contribute to the observed reduced size and lower yield of XDP COs. Together, these findings indicate that COs carrying the XDP-SVA exhibit early neurodevelopmental abnormalities, including impaired rosette architecture, expanded progenitor zones encompassing both radial glia and intermediate progenitors, and and elevated rate of apoptosis.

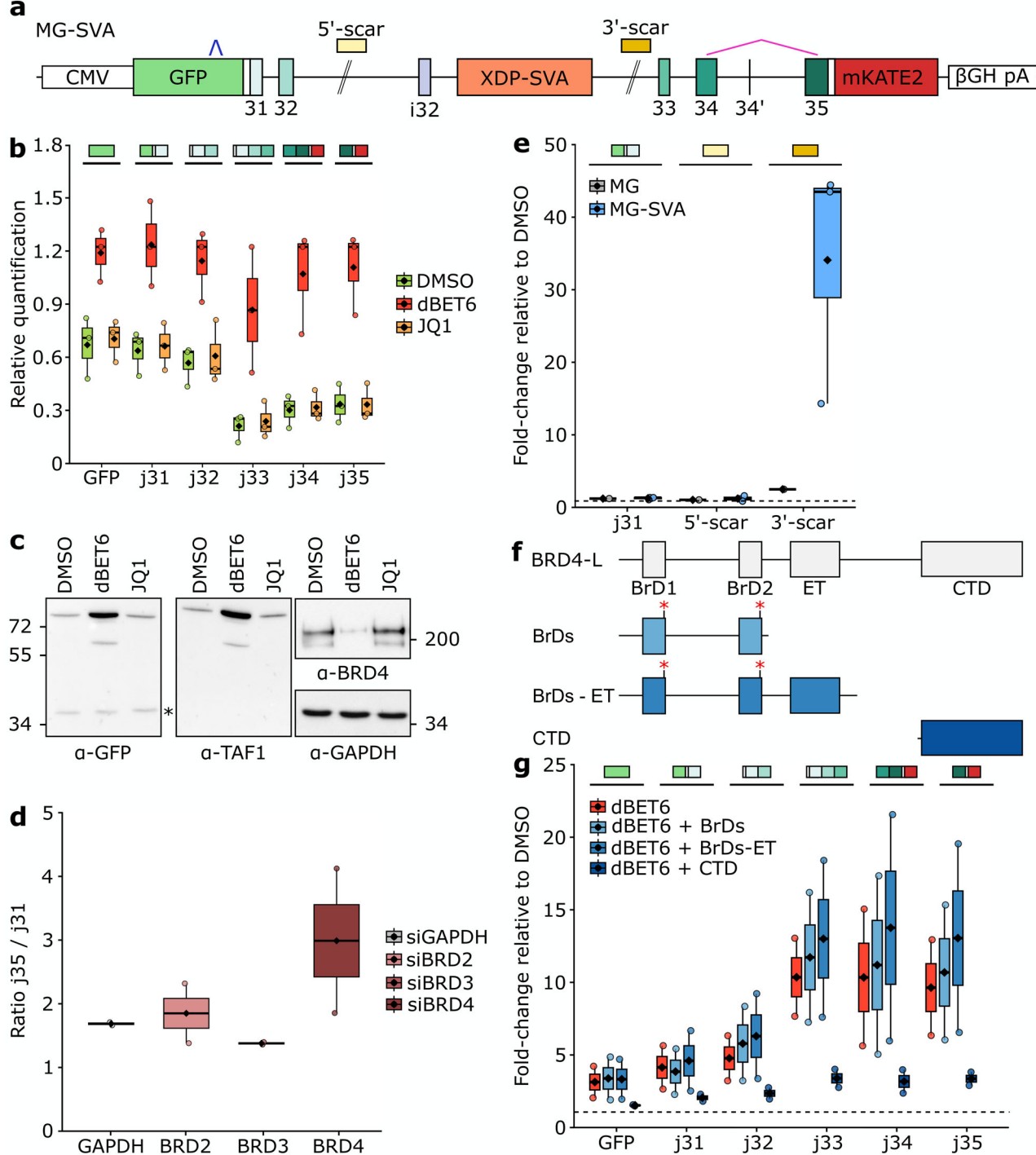

**Fig. 3 | BRD4 bromodomain-independent function links transcription regulation with 3′ RNA processing. a** Schematic of the dual-color *TAF1* minigene XDP-SVA reporter, as in Fig. 1, indicating 5′-scar and 3′-scar amplicons used in panel **e**. Effect of DMSO, BET depletion, via dBET6, versus bromodomain inhibition, via JQ1, on *TAF1* minigene transcription (**b** - box plots generated with $n = 3$ biological replicates) and the respective protein product (**c**). Truncated products are indicated with an asterisk, GAPDH serves as a BET depletion-insensitive loading control. Molecular weight is expressed in kDa. **d** Exon 35-mKATE2 to linker-exon 31 ratio in the XDP-SVA reporter line upon siRNA-mediated downregulation of *BRD2*, *BRD3*, and *BRD4*. Box plot generated with $n = 2$ biological replicates and scaled to non-targeting siRNA control. **e** RT-qPCR on chromatin-associated RNA. Box plots

generated with $n = 3$ biological replicates for the XDP-SVA containing line, while $n = 1$ for the wild-type control line. **f** dBET6-insensitive BRD4 truncations are indicated. BRD4-L (long isoform) is depicted to represent BRD4 functional domains, where: BrD1: bromodomain 1, BrD2: bromodomain 2, ET: extraterminal domain, CTD: C-terminal domain. Asterisks (*) on top of the two bromodomains indicate the two Asparagine mutations that make BRD4 fragments insensitive to dBET6-mediated degradation. **g** RT-qPCR quantification of *TAF1* minigene amplicons and their respective exon junctions upon reintroduction of BRD4 fragments. Box plots generated with $n = 2$ biological replicates. Source data are provided as a Source Data file.

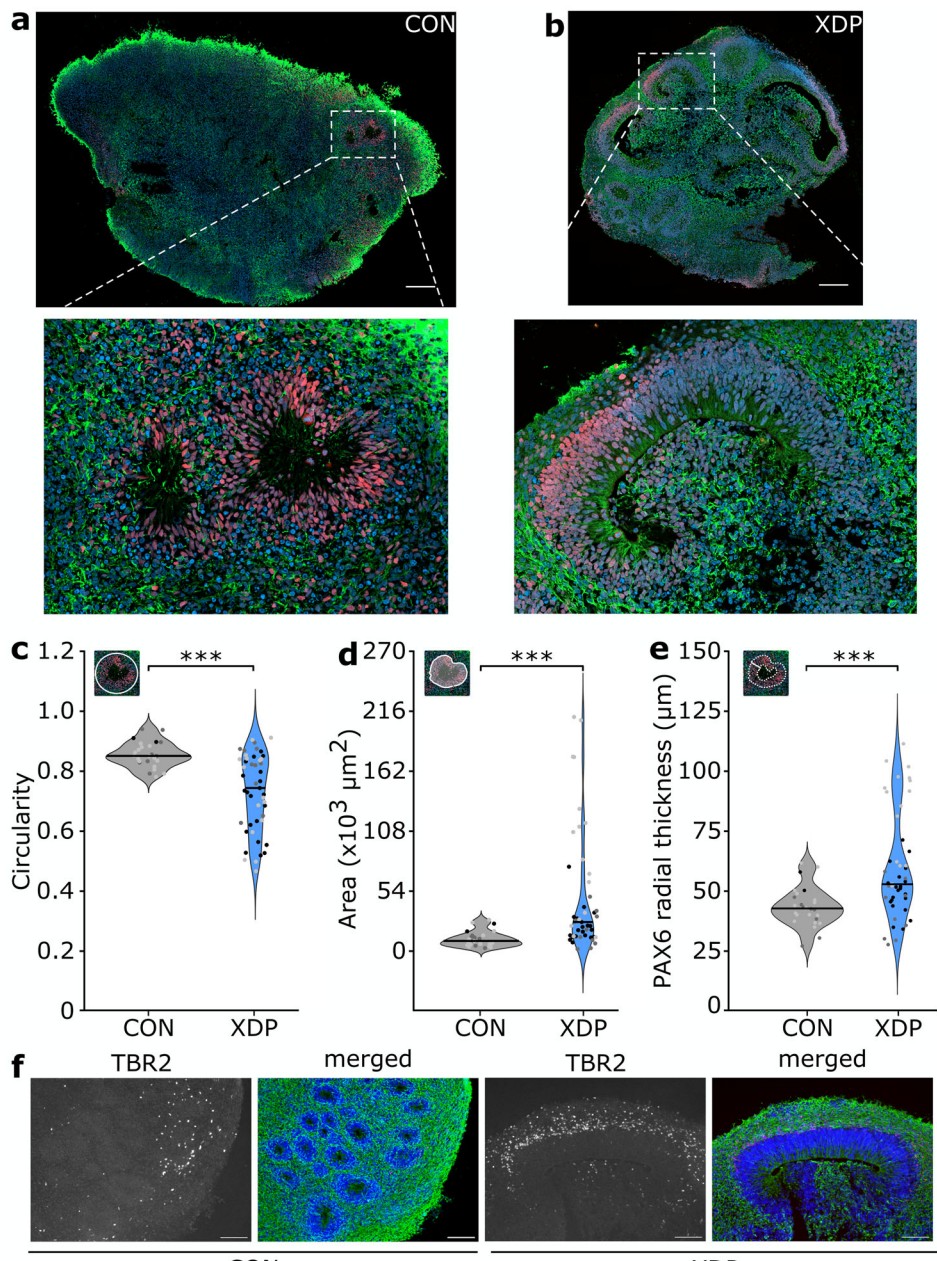

**Fig. 4 | Aberrant neuroepithelial architecture in XDP cerebral organoids.**
**a**, **b** Representative immunostaining of iPSC-derived control and XDP COs. TUJ1 (pan-neuronal marker) is shown in green, PAX6 (neural progenitor marker) in red, and nuclei are counterstained with DAPI. Scale bar: 200 μm. Quantification of neuroepithelial architecture: circularity factor (**c**), rosette area including the ventricular space (**d**), and PAX6⁺ layer thickness (**e**) in control and XDP COs. Violin plots represent pooled measurements from three biological replicates (indicated in black, dark gray and light gray) using independent iPSC lines for both genotypes. Adjusted *p*-value in (**c**) = 4.13e-05; in (**d**) = 2.77e-06, in (**e**) = 3.06e-07.
**f** Immunostaining for TBR2. Left panels show TBR2 channel alone; right panels show merged channels (TBR2 in red, TUJ1 in green, DAPI in blue). Scale bar: 100 μm. Source data are provided as a Source Data file.

## dBET6 treatment rescues the XDP molecular signature in COs

To determine whether structural abnormalities observed in XDP COs are underpinned by the characteristic *TAF1* mRNA misprocessing, we examined the fidelity of the XDP molecular signature. Using 3' RACE coupled with long-read sequencing, we identified a reproducible set of XDP-specific alternative transcripts characterized by premature termination. The most prominent was inclusion of intronic exon i32, accompanied by additional isoforms using cryptic intronic 3' splice acceptors located approximately 4.7 kb and 9.8 kb downstream of exon 32 (Fig. 5a–d). These events match cryptic acceptor sites previously annotated in XDP iPSCs[8]. These observations suggest that the presence of the XDP-SVA triggers misprocessing at various sites within intron 32 resulting in the generation of a number of prematurely terminating transcripts, which collectively decrease the proportion of full-length *TAF1* mRNAs. While the XDP signature seems consistent across pluripotent and immature neuronal cell types, it is unclear whether the same blueprint is detectable in terminally differentiated neuronal samples. Therefore, we applied the TAF1 3' RACE protocol on six *postmortem* XDP pre-frontal cortex specimens, which confirmed intronic splicing and PCPA at 4.7 kb, 9.8 kb and i32 (Fig. 5e–f).

Taken together, this describes an XDP-SVA specific subset of stable, polyadenylated, truncated *TAF1* isoforms and suggests that this molecular signature is maintained independently of neuronal

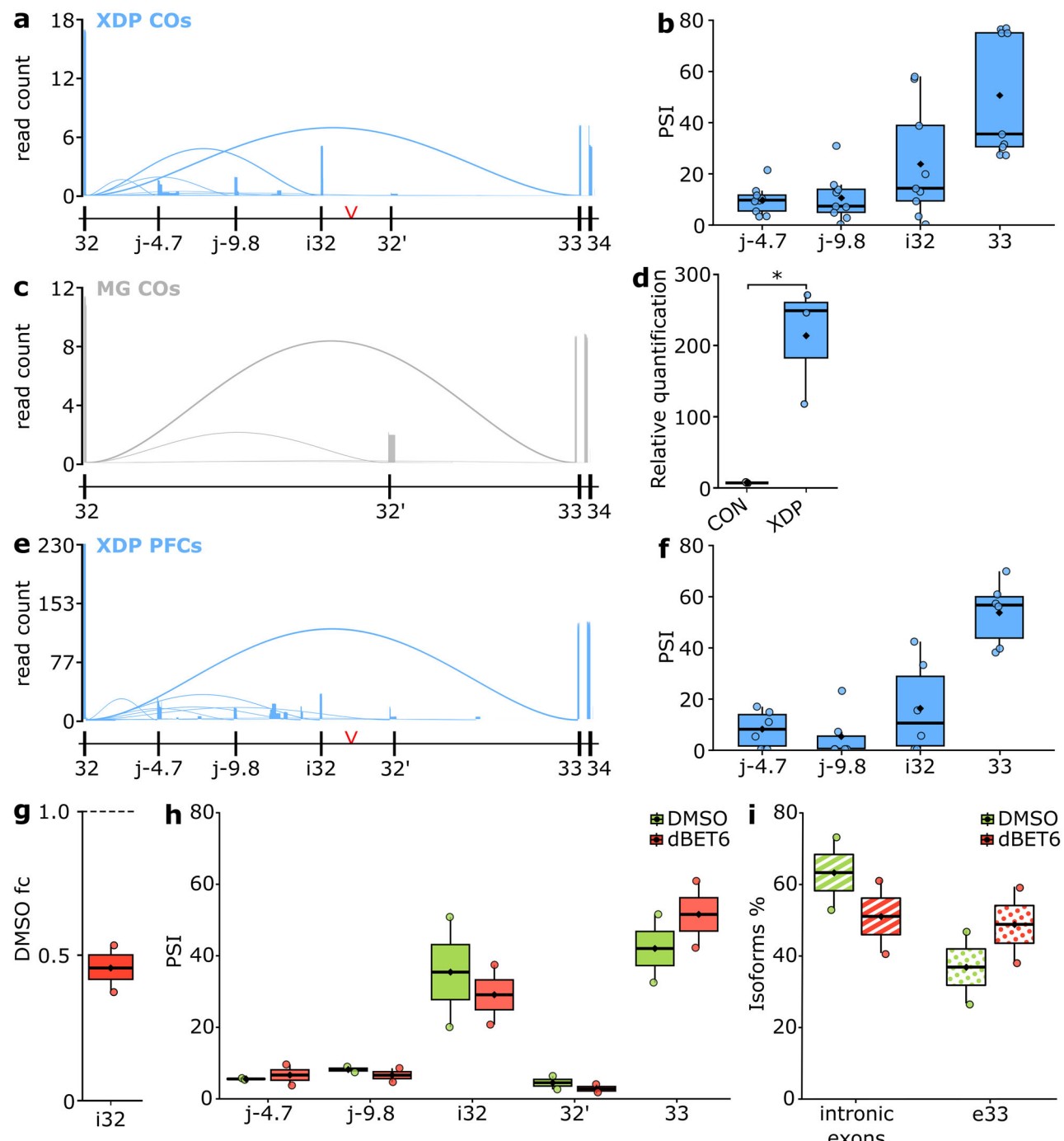

**Fig. 5 | BRD4 depletion restores physiological splicing patterns in XDP COs.**
**a** Aggregated sashimi plots showing alternative splicing events downstream of exon 32 in XDP COs. Read count is depicted as read counts/1000 (**b**) Percent-spliced-in (PSI) of acceptor sites downstream of exon 32 in XDP COs, $n = 9$ ($n = 3$ technical replicates for $n = 3$ biological replicates). **c** Aggregated sashimi plots showing alternative splicing events downstream of exon 32 in control COs. Read count is depicted as read counts/1000 (**d**) Quantification of i32 expression ($n = 3$ biological replicates per genotype; $p = 0.046$). **e** Aggregated Sashimi plot of six XDP pre-frontal cortex (PFC) specimens showing the alternative splicing events

originating from exon 32. The y-axis has been zoomed in to allow better visualization. Junctions j-4.7 and j-9.8 refer to acceptors -4.7 kb and -9.8 kb into intron 32, respectively. **f** PSI of acceptor sites downstream of exon 32 in XDP PFC specimens ($n = 6$ independent XDP patient specimens). **g** DMSO-normalized quantification of i32 expression upon dBET6 treatment. **h** PSI of acceptor sites downstream of exon 32 upon dBET6 treatment. **i** Percentage of *TAF1* transcripts containing cryptic intronic exons (patterned with stripes) or canonical exon 33 (patterned with dots) upon dBET6 treatment. For panels **g**–**i** $n = 2$ biological replicates. Source data are provided as a Source Data file.

differentiation stage and is detectable within a heterogeneous neuronal environment.

Next, we evaluated whether BRD4 depletion could rescue *TAF1* misprocessing in XDP COs. Treatment of XDP COs with dBET6 (Supplementary Fig. 6g) led to a reduction in transcripts containing exon i32 compared to DMSO-treated COs (Fig. 5g).

Additionally, we observed a downregulation in the usage of intronic 3' splice acceptor sites and an increase in the usage of the canonical exon 33 (Fig. 5h). This resulted in a modest but consistent shift in the proportion of *TAF1* isoforms that would include intronic exons versus the isoforms that would extend to exon 33 (Fig. 5i).

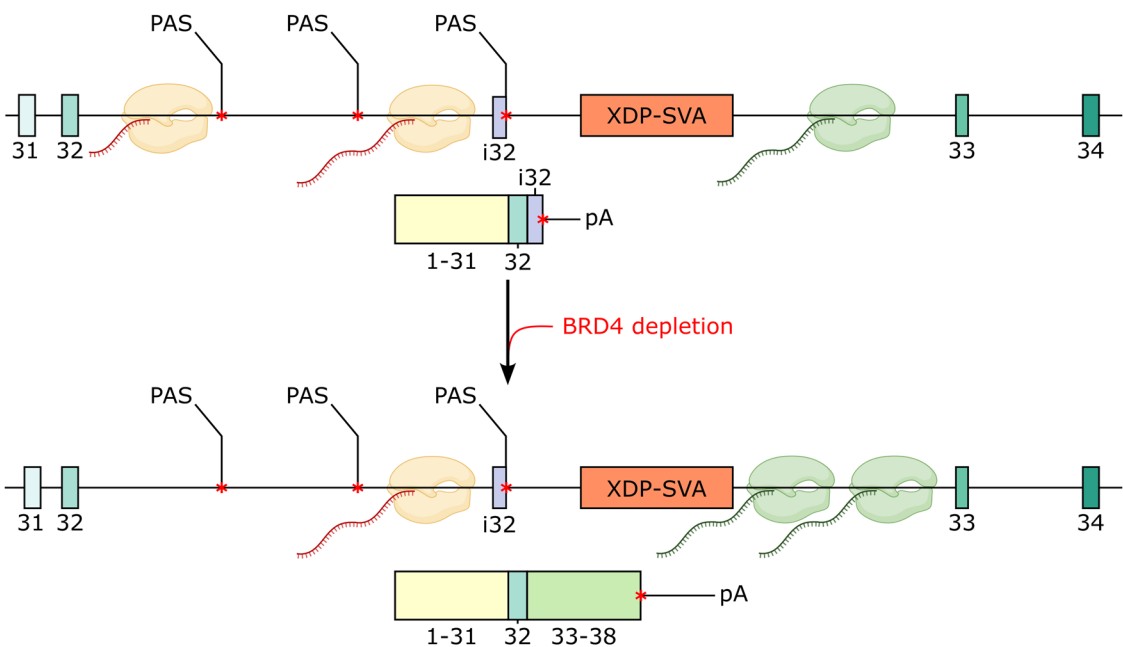

**Fig. 6 | BRD4 depletion restores mRNA processing across the XDP-SVA.** Working model depicting how BRD4 depletion reduces the XDP-SVA-induced usage of premature polyadenylation signal (PAS) and promotes transcription read-through over the *TAF1* locus. Terminating pol II molecules are depicted in yellow, while elongating molecules downstream the XDP-SVA are depicted in green. pA = polyA tail. The red asterisks indicate stop codons. The cartoons of pol II and its associated RNA has been created in BioRender. Capponi, S. (2026) https://BioRender.com/0xpirv6.

Together, these results demonstrate that BRD4 depletion regulates mRNA processing mechanisms triggered by the XDP-SVA within *TAF1* intron 32, promoting the recovery of full-length *TAF1* transcripts in a disease-relevant, 3D human model.

## Discussion

This study integrates disease modeling with targeted chemical screening to uncover the roots of the molecular mechanism for XDP. By using a dual-color reporter system, we demonstrate that the XDP-SVA induces activation of cryptic exons resulting from mis-splicing and premature cleavage and polyadenylation (PCPA) within intron 32 of *TAF1* and generation of truncated *TAF1* isoforms. Importantly, this mechanism is counteracted by depletion of the transcription regulator BRD4, in a bromodomain-independent manner. This effect was recapitulated in a patient-derived complex neuronal model reinforcing the relevance of the molecular signature and its rescue by PROTAC-mediated BET degradation. This finding mechanistically links *TAF1* mRNA misprocessing to intronic splicing and PCPA and supports transcriptional processivity as a node for therapeutic intervention in XDP.

While the *TAF1* reporter system lacks the full regulatory context of the *TAF1* locus, its ability to recapitulate the XDP molecular signature highlights the intrinsic capacity of the XDP-SVA to induce defective mRNA processing. Whether this susceptibility is entirely encoded in the sequence and structure of the XDP-SVA or is shaped by its genomic context, remains an important question for future studies. Notably, our data demonstrate for the first time that despite being driven by strong *cis*-regulatory mechanisms this aberrant processing can be reversed, prompting us to investigate how BRD4 depletion mediates this rescue. BRD4 has emerged as a multifunctional regulator of gene expression with functions beyond its canonical role as chromatin reader and transcriptional coactivator extending into mRNA processing and transcription termination. A recent study implicated BRD4 in coordinating co-transcriptional RNA maturation and in modulating polyadenylation site choice, suggesting a broader influence on 3′ end processing than previously appreciated[18]. Our findings support a bromodomain-independent role of BRD4 in alleviating premature termination, which is consistent with observations that BRD4 can engage in transcription elongation control via its C-terminal domain independent of its bromodomains[28]. The selective rescue of the XDP molecular phenotype by BRD4 depletion and not by bromodomain inhibitors underscores the importance of BRD4 non-canonical activities, which would be mediated by its CTD. As transcription termination gains attention, our work reinforces the role of BRD4 as a central integrator of chromatin architecture, elongation dynamics, and mRNA maturation.

XDP transcriptional defects are evident at the XDP-SVA insertion site in intron 32 of *TAF1*, where multiple regulatory disturbances converge. Our data support a model, in which co-transcriptional mRNA processing of intron 32 is disrupted in a subset of pol II transcription elongation complexes with the contribution of distinct layers of genetic, epigenetic and transcriptional interference. First, it has been proposed, that the genomic sequence of the XDP-SVA favors the formation of non-canonical DNA structures like G4-quadruplexes at both the variable number tandem repeat (VNTR)[29] and the hexamer repeat[30] region. These are bound by Zinc finger protein 91 (ZNF91), which is a factor involved in limiting the transcriptional activation of SVAs[31]. In this context stabilization of G4 quadruplexes exacerbates SVA-induced misprocessing of *TAF1* mRNAs and conversely, G4 destabilization alleviates it[29,30]. In parallel, the XDP-SVA is epigenetically silenced by repressive chromatin marks such as H3K9me3 as well as DNA methylation[32]. Furthermore, recent studies have revealed the presence of XDP-SVA-derived antisense RNAs forming R-loops[27], which are DNA-RNA hybrid structures with the potential to disturb elongating pol II complexes[33]. We propose that these combined regulatory disturbances collectively promote defective mRNA processing turning intron 32 into a potent termination checkpoint, which can be bypassed by modulating termination dynamics through BET depletion (Fig. 6).

Although BET depletion rescues misprocessing of *TAF1* mRNAs, this should not be interpreted as evidence that BET degraders represent therapeutic candidates for XDP. Compounds such as dBET6 induce global degradation of BET proteins and thereby, they broadly

perturb transcriptional programs. This property is intentionally exploited for cytotoxicity in cancer models and BET degraders are currently being evaluated in early-stage clinical trials. In this study, BET PROTACs served as mechanistic probes that reveal a modifiable termination checkpoint within intron 32, rather than a translational strategy as long-term treatment would be precluded by BET-degraders toxicity.

The data presented in this study stimulates several future works regarding the role of BRD4 in mRNA processing, as well the pathomechanisms associated with *TAF1* mRNA misprocessing in XDP. The challenge ahead is to dissect the details of BRD4 non-canonical functions and to identify XDP-relevant interaction nodes within the mRNA processing machinery to mitigate the toxicity associated with BRD4 depletion. At the same time, this data poses a crucial question in the XDP field: is disease progression driven primarily by a reduction in functional full-length TAF1 protein or is there a more prominent role for the variety of the generated truncated TAF1 isoforms? Our data suggest that 3' mRNA processing represents a critical regulatory checkpoint in this process raising the possibility, that both quantitative and qualitative alterations in *TAF1* transcripts might contribute to XDP pathogenesis.

Another key aspect to understand XDP is determining how *TAF1* mRNA misprocessing is linked to the development of pathological features. Although the primary neurodegenerative pathology in XDP affects striatal neurons, XDP patient-derived cerebral organoids show neuroepithelial alterations. The phenotype observed may indicate cellular stress associated with the *TAF1* mRNA misprocessing. This vulnerability may reflect the fact that cerebral organoids represent a 'sensitized' developmental model, which activates stress pathways in the absence of a physiological buffering microenvironment present during in vivo development of the brain. While cerebral organoids do not model striatal degeneration, they recapitulate transcriptional dysregulation caused by the XDP-SVA. They may reveal cellular states particularly vulnerable to defects in pol II transcriptional elongation and *TAF1* mRNA processing, which may be relevant for the maintenance of striatal neurons in the adult brain.

In summary, our study identifies BRD4 depletion as a powerful modulator of transcriptional misprocessing in XDP and highlights the potential of targeting mRNA processing to overcome SVA-induced *TAF1* dysregulation. These findings expand our understanding of BRD4 biology and more broadly, they suggest that modulating elongation and termination could offer a rational approach for treating disorders rooted in co-transcriptional dysfunctions.

## Methods

### Plasmid and cell lines generation
The *TAF1* minigene reporter was cloned in a Gateway-compatible entry vector based on the single color (GFP only) design described in ref. 34. The single-color reporter with the *TAF1* promoter was generated by replacing the original CMV promoter with the 2038 bp upstream the *TAF1* mRNA cap site (hg38, chrX: 71364239- 71366276) extended into *TAF1* 5'-UTR region (last bp located in hg38, chrX: 71366314). For the dual-color versions, the mKATE2 sequence (amplified from Addgene plasmid 48345) was inserted downstream of exon 35. Gateway recombination was used to transfer the minigene into destination vectors compatible with the Flp-In system for site-specific genomic integration.

*TAF1* minigene reporters were integrated into the genome of human Flp-In T-REx cell lines HeLa (cervical carcinoma, kind gift from Dr. G. Kops, Hubrecht Institute, NL), 293 (embryonic kidney, Thermo Fisher Scientific, cat. no. R78007), and U-2 OS (osteosarcoma, kind gift from Dr. K. Haynes, Arizona State University, USA) via the Flp-In system (Thermo Fisher Scientific). Each line contains a single genomic FRT recombination site and constitutively expresses the Tet repressor enabling stable isogenic insertion of the reporter under a tetracycline-

responsive promoter. This system ensures consistent reporter expression across cell types and allows for inducible transcriptional control. Cells were cultured using DMEM high glucose (Gibco), supplemented with 10% fetal bovine serum (Gibco). Positive integrants were selected using Hygromycin B (Invitrogen) and tested for induction using doxycycline (Merck) at the concentration of 1 µg/ml.

### Expression analysis with RT-qPCR
Total RNA was isolated using the RNeasy Kit (Qiagen) and treated with TURBO DNase (Invitrogen, Thermo Fisher Scientific). cDNA was synthesized from total RNA using SuperScript III (Invitrogen, Thermo Fisher Scientific) with random hexamer primers. RT-qPCR was performed on a CFX384 Real-Time System (Bio-Rad) using iQ SYBR Green Supermix (Bio-Rad). Relative mRNA levels (Figs. 1b, 1c, 3b; Supplementary Fig. 1e, 1f, 1g) were consistently normalized to *GAPDH* and *ACTB* signals. The j35/j31 and ji32/j35 ratios (Figs. 1d, 3d, Supplementary Fig. 3d and Supplementary Fig. 5a) were calculated as internal expression ratio. Quantification in Fig. 3e is performed by internal normalization against the upstream GFP amplicon and visualized as a fold-change relative to DMSO. Quantification in Figs. 3g and 5g is visualized as relative fold-change to DMSO. In Supplementary Fig. 4d data is shown normalized against non-targeting siRNA. Amplicons used have been validated for amplification efficiency between 90 and 110%. All amplicons used to quantify minigene-derived transcripts were designed to be reporter-specific. Although the primer set detecting exon i32 could in principle also amplify endogenous *TAF1* transcripts, exon i32 expression is nearly undetectable in HeLa cells. Quantification of intronic exon i32 (Figs. 5d, 5g) was performed using a previously published TaqMan probe strategy[8] and normalized to *GUSB* mRNAs. All primers are listed in Supplementary Table 2.

### Immunoblot and antibodies
Cells were lysed in SDS-lysis buffer (4% SDS, 160 mM Tris-HCl pH 6.7, 20% glycerol, supplemented with 0.01% BFB and 50 mM DTT) and equal amounts of protein were analysed by SDS-PAGE followed by immunoblotting. For BRD4 detection, samples were separated on pre-cast gradient gels (3–8% Tris-Acetate, Invitrogen, Thermo Fisher Scientific), which allowed improved resolution of high molecular weight species. The antibody used included: GFP (Takara, cat. no. 632381, clone ID JL8, lot A8034133, 1:2000), TAF1 (from ref. 12, 1:1000), GAPDH (Millipore, cat. no. MAB374, clone ID 6C5, lot 4047566, 1:1500), BRD4 (Cell Signaling, cat. no. 13440, clone ID E2A7X, lot 10, 1:1000), Vinculin (Santa Cruz, cat. no. sc-73614, clone ID 7F9, lot H1721, 1:1000), CDK9 (Cell Signaling, cat. no. 2316, clone ID C12F7, lot 10, 1:1000), pol II Ser2-P (Cell Signaling, cat. no. 31262, clone ID E1Z3G, lot 1, 1:1000), cleaved PARP (Cell Signaling, cat. no. 9541, lot 22, 1:1000) and Caspase 3 (Cell Signaling, cat. no. 9662, lot 19, 1:500). Secondary antibodies were α-mouse-HRP conjugate or α-rabbit-HRP conjugates (Bio-Rad).

### FREpi library curation and composition
The FREpi chemical library is a manually curated collection comprising 553 unique small-molecule targeting a comprehensive spectrum of transcription and epigenetic regulators. The library was generated by merging commercially available epigenetic libraries from Cayman Chemical (Epigenetics Screening Library, cat. no: 11076) and Tocris Bioscience (Tocriscreen Epigenetics 3.0, cat. no: 7578) supplemented with additional high-potency compounds sourced from MedChem-Express, Selleck Chemicals and opnMe portal of Boehringer Ingelheim to ensure comprehensive coverage and novelty. The collection is highly focused on potency, with compounds selected based on documented $IC_{50}$ values, predominantly in the nanomolar range. To ensure quantitative rigor and reliability in high-throughput screening, the plate layout is structured to include 41 internal replicate wells for robust hit confirmation and eight inactive control molecules to establish baseline non-specific activity. The content of the library is

richly diverse and quantitatively categorized featuring key classes such as histone deacetylase inhibitors, histone lysine methyltransferases and demethylases, inhibitors of bromodomains, DNA methyltransferases, protein arginine methyltransferases and deiminases, chromatin remodelers and several kinases. The percentage distribution of the different molecular classes is depicted in Supplementary Fig. 2a and further annotated in Supplementary Table 1. The library is formatted across eight 96-well plates with compounds at 10 mM concentration in DMSO.

## High-content live-cell screening of the FREpi library and validation

HeLa cells containing the dual-color MG-SVA reporter have been seeded in 384-well glass bottom black plates (Cellvis) at the concentration of 500 cells / 100 μl. After 24 h cells have been induced with doxycycline (Merck) at the concentration of 1 μg/ml. The treatment with the FREpi compounds (final concentration of 500 nM) has been performed 24 h post-induction. Cells were imaged 18–24 h post-treatment, prior stained with Hoechst (Invitrogen, Thermo Fisher Scientific, 1 μg/ml). Each compound was tested in technical quadruplicates and each plate included a DMSO control. Live-cell imaging was performed using the Olympus ScanR high-content screening station taking nine images for each well on the GFP, mCherry (for mKATE2 imaging) and DAPI channels with the UPLSAPO 20x NA 0.75 objective. After acquisition, all images have been processed using the Olympus ScanR analysis software with AI-assisted nuclear segmentation (Supplementary Fig. 2b).

Raw well-level mKATE2 fluorescence intensity data were post-processed by applying a toxicity filter to exclude wells with fewer than 50 segmented nuclei, followed by robust per-compound replicate quality control. Outlier wells were removed using a median absolute deviation (MAD)-based z-score cutoff of 3, and replicate variability was assessed using the robust coefficient of variation (rCV%) calculated from control wells. Compounds with fewer than 3 valid replicates or rCV% above the 90th percentile of controls were flagged as unstable but not removed from the discovery phase of data analysis (Supplementary Fig. 2c). Per-compound response values were then aggregated as the median log-transformed intensity across replicates. Robust $Z$-scores (rZ) were computed relative to the global median and MAD across all compounds. Since the data were not normally distributed, hits were defined using a fixed, direction-specific threshold of $|rZ| \geq 3$, corresponding to three MADs from the median, a conservative choice to minimize false positives while retaining strong effect sizes. For visualization, replicate-level control wells for DMSO were aggregated to a single pooled point. Compounds with rZ > 5 were classified as primary hits indicating strong rescue effects, those with rZ between 3 and 5 as potential hits with moderate but consistent effects, and those with rZ < −3 as mKATE2 repressors.

For complementary validation of the robust Z-score results, raw mKATE2 fluorescence intensities from the same toxicity-filtered and QC-annotated dataset were robustly normalized within each plate using the median and MAD of plate-specific DMSO control wells, to account for plate-to-plate variability. For each compound, replicate median log-transformed intensities were then normalized to the global median of all DMSO control wells across plates and expressed as fold changes (FC) relative to DMSO. Thresholds for high-responding compounds were set at the 95th and 99th percentiles of FC values across all non-control compounds (Supplementary Fig. 2d).

For follow-up experiments, dBET6 was chosen over other top-scoring BET degraders because it produced the most reproducible BET depletion in our system. The validation of BET PROTACs (Fig. 2b–e and Supplementary Fig. 3a–d) has been performed on HeLa cells containing the XDP-SVA *TAF1* reporter. The cells were induced with doxycycline (Merck) at the concentration of 1 μg/ml for 18–24 h, treated with increasing concentration of the selected compounds (dBET6, ARV-771,

MZ1, ZHX 3-26) for 18 h and either imaged at ScanR or processed for RT-qPCR and immunoblotting as described above.

## siRNA transfection

HeLa cells expressing the dual-color *TAF1* MG-SVA reporter were first seeded and induced with doxycycline (Merck) at the concentration of 1 μg/ml. 18–24 h post-induction, cells were treated with siRNAs against *BRD2*, *BRD3*, *BRD4*, *GAPDH* and non-targeting control (Silencer Select, Invitrogen) using siRNAiMAX (Invitrogen), according to manufacturer's instructions. Samples for RT-qPCR have been collected 48 h post-transfection and processed as described in the "Expression analysis by RT-qPCR" section.

## Nascent RNA isolation and RT-qPCR

The isolation of nascent RNA was performed following the procedure published by Churchman and Mayer[35]. After cDNA conversion, the assay used in Fig. 3e targeted three *TAF1* minigene-specific regions (j31, 5'-scar and 3'-scar) and used a GFP-specific primer set for internal normalization. Primer sequences are listed in Supplementary Table 2.

## BRD4 truncation constructs and viral transduction

The BRD4 truncation constructs were generated by PCR amplification over the full-length BRD4 construct (Addgene plasmid 90005), which already contains the two bromodomain mutations p.Asn139Ala and p.Asn433Ala, which would make the constructs insensitive to dBET6 treatment. The amplicons have been designed as Gateway-compatible and recombined into a pLenti vector featuring a N-terminus 3xHA tag, derived from the Addgene plasmid 19068. The expression of the resulting tagged transgenes was constitutive under a *PGK* promoter. Lentiviruses were produced in HEK293T cells by co-transfecting the pLenti plasmids containing the specific BRD4 truncations with packaging plasmids (psPAX2, Addgene plasmid 12260, and pVSVG2, Addgene plasmid 138479) using Fugene (Promega) according to the manufacturer's protocol. Viral supernatants were collected 24- and 48-h post-transfection, filtered through a 0.45-μm filter and used either fresh or stored at −80 °C. HeLa cells containing with the XDP-SVA-containing *TAF1* reporter were seeded at 30% confluency and transduced with viral supernatant in the presence of 10 μg/ml polybrene (Merck) to enhance transduction efficiency. After 24 h, the medium was replaced, and cells were allowed to recover for an additional 48 h before selection with Puromycine (ChemCruz, 1 μg /ml). The experiments represented in Fig. 3g were performed by inducing the transduced HeLa (doxycycline, Merck, 1 μg/ml, 18–24 h) and then treating the cells with 100 nM dBET6 (Tocris) for 18 h.

## Generation, characterization and treatment of cerebral organoids

The study design and conduct complied with all relevant regulations regarding the use of human study participants and was conducted in accordance with the criteria set by the Declaration of Helsinki. Research involving human material complied with relevant ethical regulations at our institutions and was specifically evaluated and approved by the review board of Massachusetts General Hospital (Boston, MA, USA) for generation and distribution. Written informed consent for the use of donated material for research purposes was obtained from all donors. Cerebral organoids (COs) were generated from human induced pluripotent stem cells (iPSCs) following a previously published protocol[36]. Briefly, iPSCs derived from three XDP patients (32517, 35833 and 34363) and three male controls (33113, 33114 and 33362) were maintained under feeder-free conditions on Geltrex (Gibco, Thermo Fisher Scientific)-coated plates in StemFlex medium (Gibco, Thermo Fisher Scientific) and passaged using Versene (Gibco, Thermo Fisher Scientific). Reprogramming, characterization of pluripotency, confirmation of karyotype and differentiation capacity of the used iPSCs have been extensively characterized[8,37]. CO

generation was initiated using V-shaped 96-well plates, followed by neural induction and expansion phases as described[36]. COs were moved into spinning bottles, which marked day (d) 1 of the organoid culture, and harvested at d40.

For morphological analysis, cerebral organoids were fixed in 4% paraformaldehyde (PFA) at 4 °C for 45 min, washed thoroughly in PBS, and cryoprotected by immersion in 30% sucrose solution (w/v in PBS) at 4 °C until fully equilibrated. COs were then embedded in freshly prepared embedding medium (10% (w/v) sucrose, 7.5% (w/v) porcine skin gelatin in PBS), frozen on dry ice and stored at −80 °C. Cryosections were obtained using a cryostat (Leica CM3050 S) at a thickness of 10 μm at −20 °C and collected on Superfrost Plus adhesion microscope slides (Epredia). Sections were stored at −20 °C until immunostaining.

For staining, slides were equilibrated to room temperature, permeabilized in 0.2% Triton X-100 in PBS for 20 min and blocked in blocking solution (5% normal goat serum in PBS) for 1.5 h at 4 °C. Primary antibodies were diluted in blocking solution and incubated overnight at 4 °C in a humidified chamber. Primary antibodies included: PAX6 (Invitrogen, cat. no. 42-6600, lot YL388278, 1:100), TUJ1 (R&D Systems, cat. no. MAB 1195, lot HGQ0421032, 1:500) and TBR2 (Abcam, cat. no. ab23345, clone ID EPR19012, 1:500 - kind gift from Dr. S. Arnold, University of Freiburg, DE). On the next day slides were washed three times with PBS and incubated with secondary antibodies α-rabbit Alexa 568-conjugated or α-mouse Alexa 488-conjugated (Invitrogen, Thermo Fisher Scientific, 1:500) diluted in blocking solution for 1 h at room temperature. Nuclei were counterstained with DAPI (Biotiome, 1 μg/mL in PBS) for 5 min. Slides were mounted with mounting medium (Dako) and imaged.

Imaging for Fig. 4a–b was performed using a Zeiss Celldiscover 7 (CD7) automated microscope. Images were acquired using a 50×/0.5 water-immersion objective. Autofocus, tile scan settings and background correction were applied. Image acquisition parameters were kept constant across experimental conditions. Morphometric assessment was performed by imaging the slides on the EVOS M5000 Imaging System (Thermo Fisher Scientific) using a 10x magnification. Quantification of area, circularity factor and thickness of the PAX6+ zones was performed on manually segmented rosettes using a custom-made ImageJ macro. Statistical comparisons between groups were performed using a linear mixed-effects model (lmer, lmerTest package in R), with the two groups (control and XDP) as a fixed effect and each biological replicate ($n = 3$ for each group) as a random effect. Degrees of freedom and $p$-values were calculated using Satterthwaite's adjustment. Imaging for Fig. 4f and Supplementary Fig. 6c–e was performed using the EVOS M5000 Imaging System.

At d40, COs were collected for further analysis, including immunoblot and RNA analysis. Samples were processed either at physiological condition (Fig. 5a–d) or treated with dBET6 (Tocris, 100 nM, 6 h. Figures 5g–i). For each experiment, two to three individual COs were analyzed per genotype, without pooling. Each genotype represents a distinct biological replicate, and each CO was processed independently.

### Processing of post-mortem tissues
For the experiments depicted in Fig. 5e–f, RNA was isolated from six XDP brain specimens provided by CCXDP (IDs 17.01, 17.06, 17.10, 17.12, 17.13 and 17.17 – ref. [38]) as previously described[34].

### 3' RACE and long-read sequencing
3' RACE was performed using published procedure[39]. Briefly, cDNA synthesis was primed with a modified oligo(dT) primer ($Q_T$) which selectively target polyadenylated transcripts and adds two specific sequences, $Q_O$ and $Q_I$, for subsequent amplification. PCR amplification was carried with PrimeSTAR GXL DNA Polymerase (Takara Bio) using a gene-specific forward primer located at the beginning of *TAF1* exon 30 in combination with $Q_O$ reverse primer. A subsequent nested reaction

increased specificity by using an inner forward primer located at the end of *TAF1* exon 30 in combination with $Q_I$. Amplified products were purified with AMPure XP Beads (Beckman Coulter) and further processes for Nanopore long-read sequencing using the kit SQK-NBD114.24 or SLK114 combined with the barcoding kit EXP-PBC-096 according to manufacturer's instructions. Libraries run on MinION device, mounting MinION or Flongle flow cells for 24–72 h.

Data analysis for long-read sequencing was performed using a combination of Nanopore tools and custom Python scripts. Squiggles were basecalled using Dorado v0.7.1, demultiplexed using Guppy and aligned using minimap2 applying the following arguments: -ax splice -B3 -O3,6, to allow the detection of microexons. Reads were additionally filtered to require anchoring at exon 31 of *TAF1*, the presence of the $Q_I$ adapter sequence within 150 nucleotides of either read end (allowing up to one mismatch), and a poly(A) or poly(T) stretch of at least 20 nucleotides within a 150-nucleotide window. Reads from the XDP brain specimens were not filtered. Events originating from the donor splice site of *TAF1* exon 32 were extracted using custom Python scripts and aggregated and visualized using trackplot[40]. The cut-off of a minimum of 100 supported reads for each identified junction was applied. Percent spliced-in (PSI) values were calculated as the proportion of reads supporting a given splice junction relative to the total number of *TAF1* exon 32-initiated junction reads within each sample, using the following formula:

$$PSI = \left( \frac{reads_x}{\Sigma reads_{exon32}} \right)$$

The inclusion is expressed in percentage and the cut-off of 0.1% has been applied.

Transcript isoform assignment was performed using IsoQuant[41] (v3.10.0) (--data_type nanopore, --stranded none), with no restriction to annotated transcripts, allowing detection of novel splice events. To reduce inflation of isoform diversity due to truncated 3' fragments inherent to RACE libraries, an anchoring filter was applied. To obtain biologically interpretable isoform groups, exon-block patterns were collapsed into a single collapsed isoform category when they contained the same number of junctions and corresponding donor and acceptor sites matched within ±3 bp. Collapsed isoform categories supported by fewer than 100 reads were excluded from downstream quantitative analyses. Exon inclusion percentages were computed as the fraction of reads assigned to isoform categories containing a given exon or intronic exonization block, relative to the total number of reads in that sample.

### Statistics and reproducibility
All box plots show the median (center line), the 25th and 75th percentiles (box bounds), and whiskers extending to the most extreme values within 1.5x the interquartile range. Points beyond the whiskers are shown as outliers. Statistical significance in Figs. 1d, 5d and Supplementary Fig. 6b is calculated using Welch's two-sided t-test. Statistical comparisons in Figs. 4c to 4e were performed using a linear mixed-effects model and $p$-values calculated using Satterthwaite's adjustment for degrees of freedom (two-sided tests).

Data presented in Figs. 1g and 1h and Fig. 2c were independently repeated three times with similar results. Data presented in Fig. 4f was independently repeated using two XDP-derived cerebral organoids obtaining similar results.

### Reporting summary
Further information on research design is available in the Nature Portfolio Reporting Summary linked to this article.

## Data availability

Nanopore datasets have been deposited to the Sequence Read Archive (SRA) portal of the NCBI with Bioproject accession ID PRJNA1306395. Source data are provided with this paper.

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

## Acknowledgements

We thank the CCXDP biobank, in particular Dr. Ellen Penney for providing post-mortem brain specimens and the XDP patients and their unaffected relatives for participating and providing biomaterials for hiPSC generation. We thank Dr. Manfred Jung for advice on FREpi compounds, the FiPS Core Facility (T.V.), Medical Center - University

Freiburg, Germany, for assistance with the growth of human cerebral organoids, and Dr. Frank Anicet Ditengou (Bio Imaging Core Light Microscopy and Lighthouse Core Facility, University of Freiburg) for the technical assistance with CD7 imaging. We also thank the members of the Timmers lab for discussions and Timothy Chan and Laura Pulido-Cortes for comments on the manuscript. Funding: This work was supported by the CCXDP (to M.T.), the Deutsche Forschungsgemeinschaft (SFB992 to M.T. Project number A07 and A04 and TI688/1-1 to M.T) and by the Hans A. Krebs Medical Scientists Program of the Faculty of Medicine, University of Freiburg (to S.C.). Lighthouse Core Facility is supported in part by the Medical Faculty, University of Freiburg (Project Number 2021/B3Fol).

## Author contributions

S.C. and H.Th.M.T. conceived the study. S.C. designed and generated the *TAF1* reporter systems and performed molecular and cellular experiments, including compound screening, BET perturbation studies, 3' RACE, and the generation, characterization and treatment of cerebral organoids, and conducted data analysis. S.E. provided technical assistance with compound screening, immunoblot analyses of BET perturbation experiments and staining of cerebral organoids. Z.C. assisted with RT-qPCR experiments. F.G. and T.V. supervised and guided the generation of cerebral organoids. C.A.V. and D.C.B. provided XDP and control hiPSCs and assisted with cell culture. E.Ö.-G. assembled the FREpi library. M.F. provided guidance for the FREpi library screening. S.C. wrote the manuscript with input from all authors. S.C. and H.Th.M.T. acquired funding. All authors reviewed and approved the final manuscript.

## Funding

## Competing interests

The authors declare no competing interests.
