## [Transparent Peer Review file · Nature Communications]

Correction of the molecular phenotype of X-linked dystonia-parkinsonism reveals a non-canonical function of BRD4

Corresponding Author: Professor Marc Timmers

Version 0:

Reviewer comments:

Reviewer #1

(Remarks to the Author)

Capponi and colleagues present a report on the molecular pathogenesis of X-linked dystonia-parkinsonism syndrome (XDP). This rare disease is caused by the insertion of a SVA element in intron 32 of the TAF1 gene. The findings offer compelling evidence of altered splicing in multiple cell lines, with the presence of shorter isoforms of the TAF1 gene. The screening of compounds capable of modulating the molecular signature of the disease revealed that PROTAC compounds, specifically degraders of proteins with the Bromodomain and Extra-Terminal domain element, were able to increase the wild-type transcript. It was also determined that BRD4 is the key player in this rescue function. The molecular mechanism of BRD4 is independent of the bromodomains, but likely linked to its non-canonical transcription-related functions on the C-terminal domain. These functions allow the protein to skip the premature termination caused by the SVA insertion.

Furthermore, cerebral organoids derived from patients' induced pluripotent stem cells (iPSCs) exhibited aberrant organization and increased apoptosis. In this model, the downregulation of BRD4 was also found to be capable of restoring the canonical TAF1 transcript. The data suggests that BRD4 modulation is a promising treatment for XDP.

The paper is well-written, the results are clear and based on solid experiments. The discussion is thorough, and the materials and methods are described in detail.

The figures' legends are very detailed, which makes the results easy to understand.

The publication of this work is not precluded by any significant issues.

Minor points:

1. OMIM references for XDP and TAF1 should be added in the introduction

Reviewer #2

(Remarks to the Author)

Overall this manuscript provides interesting and compelling evidence that BRD4 regulates premature cleavage and polyadenylation (PCPA) of the TAF1 gene, leading to loss of protein. However, the manuscript is densely and poorly written, making it difficult to follow. Moreover, the description of the mechanism as "splicing" rather than alternative (intronic) polyadenylation is extremely confusing. It isn't until Figure 3 that it suddenly becomes clear that what is happening with the presence of the SVA insertion is that the message is truncated (cleaved and polyadenylated) within the SVA, resulting in loss of the rest of the transcript. As a secondary consequence of this premature cleavage, exon 32 is joined to a cryptic exon 32i as exon 33 no longer exists. This is a classic example of PCPA leading to atypical splicing.

The manuscript needs to be reframed and rewritten with this understanding, which would greatly simplify the story and clarify the mechanism. In addition, 3' RACE assays or northern blots should be done to quantify this PCPA, as qPCR with junction primers is really not appropriate in this case (qPCR is also not appropriate as a measure of alternative splicing, but now realizing that what is happening is PCPA, it is more clear why the standard methods of quantifying splicing by RT-PCR cannot be used).

Finally, more care needs to be taken in the figures and legends. For example, in many of the panels light and dark blue are used for MG vs MG+SVA, but in panel 2D these same colors are used for different junctions and no MG control is shown (this needs to be provided).

Reviewer #3

(Remarks to the Author)

This study by Capponi and colleagues presents the development of an enhanced TAF1 minigene system with SVA incorporation and a dual-fluorophore design, intended as a screening tool in the context of X-linked Dystonia-Parkinsonism (XDP). The authors also extend previously described non-canonical BRD4 mechanisms to this disease model. The mechanistic aspects build on prior studies rather than uncovering entirely new biology. For example, bromodomain-independent functions of BRD4 have been reported previously: Zheng et al. (2023) demonstrated the dispensability of the bromodomains with a critical role for the C-terminal domain, and Arnold et al. (2021) established BRD4's involvement in 3' mRNA processing. The present work advances the field by contextualizing these established mechanisms within XDP and exploring how they might be leveraged for therapeutic intervention. This translational angle is interesting, especially the idea of exploiting BRD4's functions for single-gene "rescue." However, the study places considerable emphasis on the effects of dBET6 at the TAF1 locus, drawing therapeutic conclusions primarily based on the increased restoration of the full-length transcript and protein. This leaves unaddressed the broader, genome-wide impact of BET protein degradation, which is central to evaluating both the specificity and feasibility of this approach in a therapeutic setting. Overall, the mechanistic novelty is modest, and additional work, particularly transcriptome-wide analyses and functional amelioration assays, would be crucial to fully assess the translational relevance of dBET6 for XDP. Hence, we do not consider this manuscript ready at this stage.

Major Comments

Figure-by-Figure Considerations

The manuscript describes a set of constructs that build on prior work (PMID: 34746789), now presented in an improved form with an additional C-terminal tag. While the tool idea is interesting, several aspects of the experimental design and interpretation benefit from expansion and clarification.

It is necessary to provide a clearer explanation for why the CMV promoter was chosen over the endogenous TAF1 promoter, given that both systems have been developed and tested. It is known that the CMV promoter is silenced in specific cellular contexts, such as HeLa cells, during prolonged culture. This might lead to the introduction of artifacts through the regulation of the SVA to the CMV promoter compared to the control that does not have it. Given that the authors already generated a construct under the control of the endogenous TAF1 promoter, it will add value to see additional data from this TAF1 system to support the robustness of their observations and to assess how closely the minigene recapitulates the physiological regulation of the gene.

The immunoblot data shown in Figure 1e also raise questions. If the CMV promoter enhances the intron-skipping phenotype, one would expect a graded reduction in full-length transcript accompanied by an increase in truncated isoforms, as observed in patient cells. Instead, the data reveal an almost complete loss of both full-length and truncated forms compared to the control vector, which resembles a knockout rather than the partial reduction characteristic of XDP. We were also surprised that no band is visible from the endogenous TAF1 protein, given that HeLa cells are usually expected to express it. It is necessary to clarify whether the blot was cropped or whether antibody sensitivity and/or low protein abundance could account for this result.

While a stronger effect may facilitate detection of phenotypic differences, it is not immediately clear to us how the complete absence of TAF1 would serve as a better model than reproducing the ~20% reduction observed in patients.

In Figure 2, the enrichment of BET inhibitors in the screen, together with the streamlined detection pipeline, represents an interesting experimental system. That said, the absence of details on the proprietary FREpi compound library makes it difficult to assess the robustness of these findings fully. While it is entirely understandable that individual compound identities cannot be disclosed, even a general description of pathway categories, target classes, or library diversity would help readers better appreciate whether the BET enrichment reflects a genuine biological specificity or instead a bias in the library composition.

The GFP/mKATE control ratio (~1.5) appears unexpected, as a single-transcript system would generally be expected to produce equimolar products for GFP and mKATE. It is therefore important to clarify whether this observation reflects differences in protein stability or results from another uncontrolled variable (e.g., insertion loci). Furthermore, upon treatment, the long isoform increases, while the short isoform remains relatively stable (maybe half? at most). However, this does not fully reproduce what might be expected. This pattern instead suggests that BET inhibition may primarily increase the overall amount of transcript while still leaving a considerable fraction of the truncated isoform present in the cells. From a mechanistic perspective, it remains an open question for us whether the disease phenotype in XDP is primarily the consequence of reduced full-length TAF1 protein or a dominant-negative effect exerted by the truncated isoform. Clarifying this point would substantially strengthen the interpretation of the data.

In Figure 3, the authors group the drug effects into categories intended to mimic domain mutants. The results are in line with those of Zheng et al. (2023), but we were unable to find a reference to this work in the manuscript. Furthermore, upon closer examination of the data, we noticed that the BRD2 siRNA achieved only ~60% knockdown efficiency. Given the modest yet detectable increase in the J33/J31 ratio, a more complete depletion could fulfill a complete 'rescue'. Although BRD4 knockdown produced the strongest rescue, the relatively limited efficiency of BRD2 knockdown

leaves open the possibility that a more effective depletion might yield results similar to those observed with BRD4. To address this, authors could test available BRD2 degraders or other KD strategies for the gene, which could finally lead to a more definitive comparison and clarify whether the observed effect is genuinely unique to BRD4 or shared across other members of the BET family.

Validation of BET inhibitor activity is limited to a qPCR for MYC, which is a gene sensitive to general cellular stress and changes in proliferation. Given BRD4's range of targets, it is essential to evaluate additional downstream genes, whether direct targets but ideally also those with a pre-existing SVA integration in proximity or requiring post-transcriptional regulation, similar to the mutant TAF1 minigene.

A more unbiased and broader transcriptional overview will be highly informative.

While a nascent or whole transcriptome analysis may be considered beyond the current scope, it would, for example, directly inform us on specificity and off-target risks.

Organoid-Physiology figures

The organoid utilization is, unfortunately, the part of the work we appreciated the least.

XDP is a late-onset disorder characterized by progressive striatal degeneration. The cortical organoids at day 40 used in this study predominantly contain neuronal progenitors. Morphological changes at this stage are unlikely to reflect primary pathology. If cortical neurons were central to the disease, an earlier neurodevelopmental phenotype might be expected, or clear mechanistic links between cortical function loss and striatal toxicity would need to be demonstrated first. While TAF1 splicing defects may occur broadly, cortical progenitor morphology or microcephaly of any sort is not an established hallmark of XDP, and cortical neuron loss is solely observed in very few postmortem brains, as the authors mention (cause and consequence to be then analyzed)

Established protocols and commercial kits are available for generating striatal medium spiny neurons from induced pluripotent stem cells (iPSCs) with high efficiency in 2D culture (PMID: 34385043; PMID: 36590694). Using such models provides greater disease relevance and a more direct test of dBET6 rescue potential. For instance, acute and chronic treatments with multiple doses as well as co-administration of anti-inflammatory molecules to balance the potential read-through of other SVAs in the genome, could be interesting to test.

Finally, while organoids can be valuable for drug testing, the logic of applying cortical organoids in this instance solely to show an increase in PAX6 (progenitors) and apoptotic cell number in a late-onset disease is unclear. Additionally, for the sole purpose of achieving the main result of recovering a full-length transcript, it would have been sufficient to do so in patient-derived fibroblasts or iPSCs already. If organoids are to be used, it would be more informative to evaluate whether dBET6 treatment ameliorates any morphological or functional phenotypes, not solely transcript readthrough.

The iPSC method section and supplementary information section also require additional characterization. For newly derived lines, standard pluripotency validation should be provided, as well as details on clone selection and donor matching (including age, sex, and ethnicity) should be added, at a minimum, as Extended Data. Without this information, the reported differences in phenotype may reflect line-specific variation rather than disease effects. Similarly, information on the batch reproducibility of their organoid differentiation should be clearly noted, ideally in a summary table.

Minor Comments

- Figure legends are often difficult to follow, as they frequently merge methods, interpretation, and panels. Restructuring them according to journal guidelines for clarity will improve readability.
- Extended Data Figure 1A–F presents isolated constructs without a clear connection to the main text. Better integration would strengthen the flow or remove it completely.
- Figure 3 panels should follow a logical sequence rather than prioritize visual layout.
- Extended Data Figure 3E is either missing molecular weight markers for the Pol II Ser2 blot, or the apparent ~30 kDa signal for Pol II requires clarification.
- The postmortem brain data (Figures 5a–b) are introduced relatively late in the manuscript. Moving them earlier can strengthen the justification for the construct design and locus selection, whereas their current placement is almost at the end, making them appear secondary or just a validation, and no additional information is gained at this stage.

Version 1:

Reviewer comments:

Reviewer #2

(Remarks to the Author)

I appreciate the care the authors took to address the concerns. The manuscript is much improved

Reviewer #3

(Remarks to the Author)

The authors responded to most of our comments and reasonably implemented the requested changes. The rewriting and reformulation around the molecular principles of the work clarified the nature of the study and largely resolved the impression, present in the initial submission, that this was a therapy-oriented paper. The writing is more fluid overall, figure

legends clearer, and in hindsight, we can see how some misunderstandings arose from the original presentation. We are satisfied with the revision, also in light of the two other colleagues' comments.

Two minor points:

1. We encourage authors to add, even if only briefly, a speculative sentence to the discussion on how the neuroepithelial defects observed in brain organoids (expanded PAX6+ zones, increased TBR2+ progenitors, compromised rosette architecture) could relate to the downstream striatal pathology, the primary site of neurodegeneration in XDP. At present, the link to the disease-relevant cell type is based on precedents from others' work and the reported model's suitability. A sentence orienting the general reader to how the dysregulation of such cortical progenitors could plausibly be related to striatal pathogenesis would, in our opinion, significantly strengthen the discussion and be useful to a non-expert broader audience.

2. Page 5, results section: The sentence "The TAF1 reporter assay combined with comprehensive compound screening provided mechanistic insights linking the molecular signature of XDP to mRNA alteration" appears incomplete. We suggest something like "alteration of mRNA processing" or equivalent.

General response to reviewers

The response to our manuscript of the reviewers to our manuscript is rather diverse, but we reply to each of the raised issues in a point-by-point basis. In the revised manuscript we included new experimental data in Figure 1e, 1f, 2d, 2e, 5a, 5b, 5c, 5h and 5i and in Extended Data Fig. 1f, 1g, 1h, 2d, 2e, 2f and 2g resulting from seven new experiments. The manuscript has now been restructured the manuscript. Taken together, this led to significant improvements, for which we thank the reviewers.

Additionally, previous and present work refers to the intronic exon tas i32 (PMID: 29474918, 41279153, 41282719). In order to avoid confusion in the field, we changed 32i to i32 throughout the manuscript.

Response to reviewer #1:

The paper is well-written, the results are clear and based on solid experiments. The discussion is thorough, and the materials and methods are described in detail. The figures' legends are very detailed, which makes the results easy to understand. The publication of this work is not precluded by any significant issues.

>We are thankful for the praise for our work by this reviewer. We now fixed the minor point of the OMIM references by including them into the sentence on page 2 and indicated orange.

Response to reviewer #2:

Point #1:

... the manuscript is densely and poorly written, making it difficult to follow.

>We have completely restructured the results section. We now introduce the molecular concept of the disease earlier in the text. The changes are indicated in blue.

Point #2:

Moreover, the description of the mechanism as "splicing" rather than alternative (intronic) polyadenylation is extremely confusing. It isn't until Figure 3 that is suddenly becomes clear that what is happening with the presence of the SVA insertion is that the message is truncated (cleaved and polyadenylated) within the SVA, resulting in loss of the rest of the transcript. As a secondary consequence of this premature cleavage, exon 32 is joined to a cryptic exon 32i as exon 33 no longer exists. This is a classic example of PCPA leading to atypical splicing. The manuscript needs to be reframed and rewritten with this understanding, which would greatly simplify the story and clarify the mechanism.

>We note that it is presently unclear whether cryptic splicing follows pre-mRNA cleavage and polyadenylation or *vice versa*. This mechanism has not been solved for *TAF1* mRNAs obstructed by the XDP-SVA. Our reading of the recent literature indicates that U1 binding may play an important role in suppressing premature cleavage/poly-adenylation, which indicates that intronic splicing precedes polyadenylation (PMID: 41610855, 37260513, 39632657). However, we prefer to keep this issue open as it is not essential for the interpretation of our results. Therefore, we refer to 'XDP-SVA mediated defects in proper *TAF1* mRNA processing' throughout the manuscript.

As a side note, polyadenylation does not occur within the SVA but rather upstream at the end of intronic exon 32i, which harbors the consensus AATAAA and downstream G-rich elements for polyA addition.

Point #3:

.... 3' RACE assays or northern blots should be done to quantify this PCPA, as qPCR with junction primers is really not appropriate in this case (qPCR is also not appropriate as a measure of

alternative splicing, but now realizing that what is happening is PCPA, it is more clear why the standard methods of quantifying splicing by RT-PCR cannot be used).

>We are confused by this comment as results from 3'RACE followed by long-read sequencing were included in Fig. 5. We now moved this data forward to Fig. 1e and 1f. As requested by the reviewer, we now performed the 3'-RACE on mRNA derived from HeLa cells expressing the *TAF1* minigene SVA driven by the CMV or TAF1 promoter and subjected to dBET6 treatment. Indeed, we observed the same SVA-dependent 32i inclusion into minigene-derived *TAF1* mRNAs as reported before for *TAF1* mRNAs from cell lines of XDP patients (PMID: 29474918) and as confirmed in our 3' RACE of XDP brains and XDP-hiPSCs derived cerebral organoids. In addition, the 3'RACE analyses have been extended significantly by including isoform quantifications. We included the HeLa data as Fig. 1e, 1f, 2d, 2e and 5i as Extended Data Fig. 2e, 2f and 2g. We amended the text on pages 3-7 as indicated in blue.

Point #4:

Finally, more care needs to be taken in the figures and legends. For example, in many of the panels light and dark blue are used for MG vs MG+SVA, but in panel 2D these same colors are used for different junctions and no MG control is shown (this needs to be provided).

> We apologize and thank the reviewer for carefully checking the color schemes of the figures. While we are consistent in the use of grey for the control MG and of blue for the MG+SVA, confusion was created by the color scheme of original Fig. 2D. To avoid this, we now added Extended Data Fig. 2d to overlay results from the *TAF1* mini-gene and the TAF mini-gene SVA.

Response to reviewer #3:

Point #1:

... bromodomain-independent functions of BRD4 have been reported previously: Zheng et al. (2023) demonstrated the dispensability of the bromodomains with a critical role for the C-terminal domain, and Arnold et al. (2021) established BRD4's involvement in 3' mRNA processing. The present work advances the field by contextualizing these established mechanisms within XDP and exploring how they might be leveraged for therapeutic intervention.

>We respectfully disagree with the remark that our work is only 'contextualizing these established mechanisms within XDP.' Below we stipulate the three major findings of our manuscript:

- the first demonstration of BRD4's role in SVA-mediated mRNA processing defects
- the first description of the XDP cellular phenotype in cerebral organoids.
- the first molecular correction of XDP transcriptional defects in patient-derived human models.

Point #2:

This translational angle is interesting, especially the idea of exploiting BRD4's functions for single-gene "rescue." However, the study places considerable emphasis on the effects of dBET6 at the TAF1 locus, drawing therapeutic conclusions primarily based on the increased restoration of the full-length transcript and protein. This leaves unaddressed the broader, genome-wide impact of BET protein degradation, which is central to evaluating both the specificity and feasibility of this approach in a therapeutic setting.

>We agree that BET protein degradation will not be a therapeutic option, which we did not imply in our manuscript. Toxicity of BET-PROTACs would prevent any long-term treatment, which would be required to halt progression of this neurodegenerative disease. We now explicitly state this on page 9: 'In this study, BET PROTACs served as mechanistic probes that reveal a modifiable termination checkpoint within intron 32, rather than as a translational strategy, as long-term treatment would be precluded by BET-degraders toxicity' indicated in green.

We also like to stress that our current study identifies BRD4 as a crucial component for SVA-mediated defects in mRNA processing. The identification of the CTD of BRD4 as essential for these defects provides an important way forward to avoid the toxicities associated with depletion of BRD4 and/or of loss of CDK9/cyclin T functions. Again, we stress this now on page 9: 'The challenge ahead is to dissect the details of BRD4 non-canonical functions and to identify XDP-relevant interaction nodes with the mRNA processing machinery to mitigate the toxicity associate with BRD4 depletion' indicated in green.

Point #3:

Overall, the mechanistic novelty is modest, and additional work, particularly transcriptome-wide analyses and functional amelioration assays, would be crucial to fully assess the translational relevance of dBET6 for XDP. Hence, we do not consider this manuscript ready at this stage.

>We respectfully disagree, see response to point #1 and #2.

Major Comments

Point #4:

It is necessary to provide a clearer explanation for why the CMV promoter was chosen over the endogenous TAF1 promoter, given that both systems have been developed and tested.

>We now indicate the detection advantage of CMV driven TAF1 reporter on page 4: 'To test this idea, the dual-color XDP-SVA TAF1 reporter driven by the CMV promoter provided a sensitive platform for small molecule screening based on automated microscopy to identify compounds capable of increasing mKATE2 over GFP fluorescence' indicated in green.

Point #5:

It is known that the CMV promoter is silenced in specific cellular contexts, such as HeLa cells, during prolonged culture.

>We respectfully disagree with the statement that the CMV promoter is silenced in the FRT locus in HeLa cells, or other FRT cell lines that we use regularly in our studies like U2-OS, RPE-1, HEK293 and U373. Instead, we observe that CMV-driven expression levels remain unchanged upon prolonged doxycycline-induction and at high passage numbers. Possibly, the reviewer is confused by epigenetic repression of retrovirally-transduced genes.

Point #6:

Given that the authors already generated a construct under the control of the endogenous TAF1 promoter, it will add value to see additional data from this TAF1 system to support the robustness of their observations and to assess how closely the minigene recapitulates the physiological regulation of the gene.

>To address the promoter specificity point, we show dBET6-responsiveness of the TAF1 promoter-driven constructs in Extended Data Fig. 2g. As we expected, there are no differences between the CMV and TAF1 promoters (see also Extended Data Fig. 1f and 1g), other than that the CMV promoter yields much higher expression levels of the TAF1 mini-gene as one would predict.

Point #7:

The immunoblot data shown in Figure 1e also raise questions. If the CMV promoter enhances the intron-skipping phenotype, one would expect a graded reduction in full-length transcript accompanied by an increase in truncated isoforms, as observed in patient cells. Instead, the data reveal an almost complete loss of both full-length and truncated forms compared to the control vector, which resembles a knockout rather than the partial reduction characteristic of XDP.

>We disagree with the assumption that the CMV promoter 'enhances an intron-skipping phenotype'. We identified HeLa cells as particularly suitable system recapitulating mRNA processing defects in XDP-patient derived formats like hiPSC-derived cerebral organoids and XDP brain tissue as confirmed by 3'RACE.

Point #8:

We were also surprised that no band is visible from the endogenous TAF1 protein, given that HeLa cells are usually expected to express it. It is necessary to clarify whether the blot was cropped or whether antibody sensitivity and/or low protein abundance could account for this result.

>Indeed, the endogenous 250-kDa TAF1 protein is not visible in this blot. The purpose of the experiment was to detect the mini-gene derived GFP-TAF1^{exon31/35} protein, which requires much less lysate than detection of endogenous TAF1 as a control for equal loading we used GAPDH (Fig. 1g).

Point #9:

While a stronger effect may facilitate detection of phenotypic differences, it is not immediately clear to us how the complete absence of TAF1 would serve as a better model than reproducing the ~20% reduction observed in patients.

>We disagree that TAF1 is completely absent. We showed reduced amounts of TAF1 mini-gene derived mRNAs by RT-PCR (Fig. 1b), 3'RACE (Fig. 1e) and proteins by long exposure immunoblot (Fig. 1g) by the presence of the XDP-SVA. The strong reduction of the TAF1 mini-gene expression in HeLa cells facilitated the identification of BET degraders. These hits were confirmed in a variety of other assays including XDP-patient derived cerebral organoids.

Point #10:

.. the absence of details on the proprietary FREpi compound library makes it difficult to assess the robustness of these findings fully. While it is entirely understandable that individual compound identities cannot be disclosed, even a general description of pathway categories, target classes, or library diversity would help readers better appreciate whether the BET enrichment reflects a genuine biological specificity or instead a bias in the library composition.

>We apologize for omitting a detailed description of FREpi library. This is now corrected by including Supplemental Fig. 1a and Supplemental Table 1, and by including a paragraph in the Materials and Methods on page 11 indicated in green. Dr. Özyerli-Gökner was the main driver of FREpi construction and she is now included as an author. In addition, we recognize support from Prof. Manfred Jung in FREpi design and he is now included in the acknowledgement section.

Point #11:

The GFP/mKATE control ratio (~1.5) appears unexpected, as a single-transcript system would generally be expected to produce equimolar products for GFP and mKATE. It is therefore important to clarify whether this observation reflects differences in protein stability or results from another uncontrolled variable (e.g., insertion loci). Furthermore, upon treatment, the long isoform increases, while the short isoform remains relatively stable (maybe half? at most). However, this does not fully reproduce what might be expected.

>We like to stress that the GFP and mKATE2 signals are expressed as fluorescence intensity units and they are not normalized for fluorescence yield, detector sensitivity, etc. To indicate this we now include in the legend of Fig. 2b on page 20: 'Dose-dependent effect of dBET6 on mKATE2/GFP ratio of fluorescence signals' and of Extended Data Fig. 2a, 2b and 2c on page 22: 'a-c) Dose-dependent effect of ARV-771 (a), MZ1 (b) and ZXH-3-26 (c) on mKATE2/GFP ratio of fluorescence signals.'

Point #12:

This pattern instead suggests that BET inhibition may primarily increase the overall amount of transcript while still leaving a considerable fraction of the truncated isoform present in the cells. From a mechanistic perspective, it remains an open question for us whether the disease phenotype in XDP is primarily the consequence of reduced full-length TAF1 protein or a dominant-negative effect exerted by the truncated isoform. Clarifying this point would substantially strengthen the interpretation of the data.

>This comment touches upon an important question in the XDP field, which is a subject of intense study in multiple laboratories. However, solving this question has not been the aim of our study, which focused on development of the *TAF1* mini-gene and its application to identify small molecules capable of reversing the molecular phenotype of SVA-mediated defects in mRNA processing in XDP-relevant model systems.

Point #13:

In Figure 3, the authors group the drug effects into categories intended to mimic domain mutants. The results are in line with those of Zheng et al. (2023), but we were unable to find a reference to this work in the manuscript.

>We are confused by this comment as the Zheng study (PMID: 37442129) does not examine BRD4 effects on mRNA processing. Thus, the reference is irrelevant for drug effects observed in our study.

Point #14:

... we noticed that the BRD2 siRNA achieved only ~60% knockdown efficiency. Given the modest yet detectable increase in the J33/J31 ratio, a more complete depletion could fulfill a complete 'rescue'. Although BRD4 knockdown produced the strongest rescue, the relatively limited efficiency of BRD2 knockdown leaves open the possibility that a more effective depletion might yield results similar to those observed with BRD4. To address this, authors

could test available BRD2 degraders or other KD strategies for the gene, which could finally lead to a more definitive comparison and clarify whether the observed effect is genuinely unique to BRD4 or shared across other members of the BET family.

>Indeed, despite several efforts we have been unable to achieve a more efficient knockdown of BRD2 mRNAs than 60%. Based on the siRNA experiment we cannot exclude that BRD2 contributes to SVA-mediated repression of full length TAF1 mRNAs. The only BRD2 PROTAC reported has low potency and specificity (PMID: 40792622). However, the result of Fig. 3g shows that the CTD of BRD4 can fully reverse dBET6 release of SVA-repression. BRD2 does not contain a CTD like BRD4.

In addition, a recent study of the Shilatifard lab (PMID: 41478281) indicated that BRD2 (and BRD3) act at the level of transcription initiation rather than mRNA processing. They also reaffirm that BRD4 acts at steps later in the transcription cycle.

Point #15:

Validation of BET inhibitor activity is limited to a qPCR for MYC, which is a gene sensitive to general cellular stress and changes in proliferation. Given BRD4's range of targets, it is essential to evaluate additional downstream genes, whether direct targets but ideally also those with a pre-existing SVA integration in proximity or requiring post-transcriptional regulation, similar to the mutant TAF1 minigene. A more unbiased and broader transcriptional overview will be highly informative. While a nascent or whole transcriptome analysis may be considered beyond the current scope, it would, for example, directly inform us on specificity and off-target risks.

>Indeed, a whole transcriptome analysis does not make sense, but we would also like to point out that since 2011 MYC is known to be a primary target of BET inhibitors (PMID: 21964340).

Point #16:

The organoid utilization is, unfortunately, the part of the work we appreciated the least. XDP is a late-onset disorder characterized by progressive striatal degeneration. The cortical organoids at day 40 used in this study predominantly contain neuronal progenitors. Morphological changes at this stage are unlikely to reflect primary pathology. If cortical neurons were central to the disease, an earlier neurodevelopmental phenotype might be expected, or clear mechanistic links between cortical function loss and striatal toxicity would need to be demonstrated first. While TAF1 splicing defects may occur broadly, cortical progenitor morphology or microcephaly of any sort is not an established hallmark of XDP, and cortical neuron loss is solely observed in very few postmortem brains, as the authors mention (cause and consequence to be then analyzed).

>We like to point that the system used are cerebral organoids and not cortical organoids as assumed. In addition, we did not aim to reconstruct primary XDP pathology, which would be beyond the scope of our study. Although we agree that XDP is late on-set, we pointed out that cerebral organoids are important tools for other late on-set disorders like Parkinson's and Huntington's disease as we mentioned on page 5. We speculate that the *in vitro* growth conditions of COs aggravate *in vivo* phenotypes as indicated by Bhaduri et al (PMID: 31996853).

Point #17:

Established protocols and commercial kits are available for generating striatal medium spiny neurons from induced pluripotent stem cells (iPSCs) with high efficiency in 2D culture (PMID: 34385043; PMID: 36590694). Using such models provides greater disease relevance and a more direct test of dBET6 rescue potential. For instance, acute and chronic treatments with multiple doses as well as co-administration of anti-inflammatory molecules to balance the potential read-through of other SVAs in the genome, could be interesting to test.

>We agree that it would interesting to examine dBET6 effects on iPSC-derived striatal medium spiny neurons, but the molecular signature strength is inversely correlated with neuronal

differentiation indicating that striatal medium spiny neurons are not a good system for dBET6 effects (PMID: 29474918, 40540399). In addition, toxicity of dBET6 treatments is limiting their use for acute and chronic treatment. We do not share the expectation that other SVAs would affect transcriptional elongation. In fact, another SVA element is located at the 3'-end of intron 32 is present in the human population, which does not impair elongation.

Point #18:

Finally, while organoids can be valuable for drug testing, the logic of applying cortical organoids in this instance solely to show an increase in PAX6 (progenitors) and apoptotic cell number in a late-onset disease is unclear. Additionally, for the sole purpose of achieving the main result of recovering a full-length transcript, it would have been sufficient to do so in patient-derived fibroblasts or iPSCs already. If organoids are to be used, it would be more informative to evaluate whether dBET6 treatment ameliorates any morphological or functional phenotypes, not solely transcript readthrough.

>We respectfully repeat that the organoid system used are cerebral and not cortical organoids. Given their cellular complexity we prefer organoids over XDP patient-derived fibroblasts or iPSCs. Importantly, we observe rescue by short dBET6 treatments (Fig. 5h and 5i).

Point #19:

The iPSC method section and supplementary information section also require additional characterization. For newly derived lines, standard pluripotency validation should be provided, as well as details on clone selection and donor matching (including age, sex, and ethnicity) should be added, at a minimum, as Extended Data. Without this information, the reported differences in phenotype may reflect line-specific variation rather than disease effects. Similarly, information on the batch reproducibility of their organoid differentiation should be clearly noted, ideally in a summary table.

-We note that we are not the first to use these iPSC lines. In fact, they are well characterized (PMID: 26769797, 29474918) as now indicated in the Materials and Methods section on page 13. Importantly, they have been used widely in several publications by the XDP community (PMID: 26769797, 29474918, 40540399, 38834915, 39287133). Obviously, XDP iPSCs are derived from men of Filipino descent.

Minor Comments

Point #20:

Figure legends are often difficult to follow, as they frequently merge methods, interpretation, and panels. Restructuring them according to journal guidelines for clarity will improve readability.

-We now structured the figure legends to the journal guidelines.

Point #21:

Extended Data Figure 1A–F presents isolated constructs without a clear connection to the main text. Better integration would strengthen the flow or remove it completely.

>We respectfully disagree as these panels provide an orthogonal validation of the 3'RACE data.

Point #22:

Figure 3 panels should follow a logical sequence rather than prioritize visual layout.

-We thank the reviewer for pointing this out. We swapped the location of panel d and panel e.

Point #23:

Extended Data Figure 3E is either missing molecular weight markers for the Pol II Ser2 blot, or the apparent ~30 kDa signal for Pol II requires clarification.

>We are confused by this comment as the position of the 200-kDa marker is indicated to the right of the panel e of Extended Data Fig. 3.

Point #24:

The postmortem brain data (Figures 5a–b) are introduced relatively late in the manuscript. Moving them earlier can strengthen the justification for the construct design and locus selection, whereas their current placement is almost at the end, making them appear secondary or just a validation, and no additional information is gained at this stage.

>We thank the reviewer for this comment and we detail the justification for our design on page 3: 'The construct spans the human TAF1 locus from exon 31 to 35 and incorporates the XDP-SVA at its native landing position within intron 32, which has been reconstructed using the native 5' and 3' regions and retaining ~1 kb of intronic sequence flanking the XDP-SVA, including the previously described cryptic exon i32⁸'.

Point-by-point response to reviewer #3

We thank the reviewer for the careful re-evaluation of the manuscript.

“1. We encourage authors to add, even if only briefly, a speculative sentence to the discussion on how the neuroepithelial defects observed in brain organoids (expanded PAX6+ zones, increased TBR2+ progenitors, compromised rosette architecture) could relate to the downstream striatal pathology, the primary site of neurodegeneration in XDP. At present, the link to the disease-relevant cell type is based on precedents from others' work and the reported model's suitability. A sentence orienting the general reader to how the dysregulation of such cortical progenitors could plausibly be related to striatal pathogenesis would, in our opinion, significantly strengthen the discussion and be useful to a non-expert broader audience.”

We agree with the reviewer and we added a short paragraph on the phenotype observed in the cerebral organoids in the 'Discussion' section at page 9, highlighted in **green**.

“2. Page 5, results section: The sentence “The TAF1 reporter assay combined with comprehensive compound screening provided mechanistic insights linking the molecular signature of XDP to mRNA alteration” appears incomplete. We suggest something like “alteration of mRNA processing” or equivalent.”

We modified the sentence as suggested. The amend is reported in **green**.